# Optimal information gain at the onset of habituation to repeated stimuli

Giorgio Nicoletti[1,2,3], Matteo Bruzzone[4,5,6], Samir Suweis[3,5], Marco dal Maschio[4,5], Daniel Maria Busiello[3,7]*

[1]ECHO Laboratory, École Polytechnique Fédérale de Lausanne, Lausanne, Switzerland; [2]Quantitative Life Sciences section, The Abdus Salam International Center for Theoretical Physics (ICTP), Trieste, Italy; [3]Department of Physics and Astronomy "Galileo Galilei", University of Padova, Padova, Italy; [4]Department of Biomedical Science, University of Padova, Padova, Italy; [5]Padova Neuroscience Center, University of Padova, Padova, Italy; [6]Department of Biology, University of Fribourg, Fribourg, Switzerland; [7]Max Planck Institute for the Physics of Complex Systems, Dresden, Germany

## eLife Assessment

This manuscript presents a **valuable** minimal model of habituation which is quantified by information theoretic measures. The results here could be of use in interpreting habituation behavior in a range of biological systems. The evidence presented is **solid**, and uses simulations of the minimal model to recapitulate several hallmarks of habituation from a simple model.

*For correspondence:
busiello@pks.mpg.de

Competing interest: The authors declare that no competing interests exist.

**Abstract** Biological and living systems process information across spatiotemporal scales, exhibiting the hallmark ability to constantly modulate their behavior to ever-changing and complex environments. In the presence of repeated stimuli, a distinctive response is the progressive reduction of the activity at both sensory and molecular levels, known as habituation. In this work, we solve a minimal microscopic model devoid of biological details, where habituation to an external signal is driven by negative feedback provided by a slow storage mechanism. We show that our model recapitulates the main features of habituation, such as spontaneous recovery, potentiation, subliminal accumulation, and input sensitivity. Crucially, our approach enables a complete characterization of the stochastic dynamics, allowing us to compute how much information the system encodes on the input signal. We find that an intermediate level of habituation is associated with a steep increase in information. In particular, we are able to characterize this region of maximal information gain in terms of an optimal trade-off between information and energy consumption. We test our dynamical predictions against experimentally recorded neural responses in a zebrafish larva subjected to repeated looming stimulations, showing that our model captures the main components of the observed neural habituation. Our work makes a fundamental step towards uncovering the functional mechanisms that shape habituation in biological systems from an information-theoretic and thermodynamic perspective.

## Introduction

Sensing mechanisms in biological systems span a wide range of temporal and spatial scales, from cellular to multi-cellular level, forming the basis for decision-making and the optimization of limited resources (*Tkačik and Bialek, 2016*; *Azeloglu and Iyengar, 2015*; *Gnesotto et al., 2018*; *Whiteley et al., 2017*; *Perkins and Swain, 2009*). Emergent macroscopic phenomena such as adaptation and

habituation reflect the ability of living systems to effectively process the information they collect from their noisy environment (*Nemenman, 2012*; *Nakajima, 2015*; *Koshland et al., 1982*). Prominent examples include the modulation of flagellar motion operated by bacteria according to changes in the local nutrient concentration (*Tu et al., 2008*; *Tu, 2008*; *Mattingly et al., 2021*), the regulation of immune responses through feedback mechanisms (*Cheong et al., 2011*; *Wajant et al., 2003*), the progressive reduction of neural activity in response to repeated looming stimulation (*Marquez-Legorreta et al., 2022*; *Fotowat and Engert, 2023*), and the maintenance of high sensitivity in varying environments for olfactory or visual sensing in mammalian neurons (*Lan et al., 2012*; *Menini, 1999*; *Kohn, 2007*; *Lesica et al., 2007*; *Benucci et al., 2013*).

In the last decade, advances in experimental techniques fostered the quest for the core biochemical mechanisms governing information processing. Simultaneous recordings of hundreds of biological signals made it possible to infer distinctive features directly from data (*Schneidman et al., 2006*; *Tkačik et al., 2014*; *Kurtz et al., 2015*; *Tunstrøm et al., 2013*). However, many of these approaches fall short of describing the connection between observed behaviors and underlying microscopic drivers (*Nicoletti and Busiello, 2021*; *Nicoletti and Busiello, 2022a*; *De Smet and Marchal, 2010*; *Nicoletti et al., 2022b*). To fill this gap, several works focused on the architecture of specific signaling networks, from tumor necrosis factor (*Cheong et al., 2011*; *Wajant et al., 2003*) to chemotaxis (*Tu et al., 2008*; *Celani et al., 2011*), highlighting the essential structural ingredients for their efficient functioning. An observation shared by most of these studies is the key role of a negative feedback mechanism to induce emergent adaptive responses (*Kollmann et al., 2005*; *De Ronde et al., 2010*; *Selimkhanov et al., 2014*; *Barkai and Leibler, 1997*). Moreover, any information-processing system, biological or not, must obey information-thermodynamic laws that prescribe the necessity of a storage mechanism (*Parrondo et al., 2015*). This is an unavoidable feature highlighted in numerous chemical signaling networks (*Tu et al., 2008*; *Kollmann et al., 2005*) and biochemical realizations of Maxwell Demons (*Flatt et al., 2023*; *Bilancioni et al., 2023*). As the storage of information during processing generally requires energy (*Bennett, 1982*; *Sagawa and Ueda, 2009*), sensing mechanisms have to take place out of equilibrium (*Gnesotto et al., 2018*; *Hartich et al., 2015*; *Skoge et al., 2013*; *Lestas et al., 2010*). Recently, the discovery of memory molecules (*Coultrap and Bayer, 2012*; *Frankland and Josselyn, 2016*; *Lisman et al., 2002*) hinted at the possibility that storing mechanisms might be instantiated directly at the molecular scale. Overall, negative feedback, storage, and out-of-equilibrium conditions seem to be necessary requirements for a system to process environmental information and act accordingly. To quantify the performance of a biological information-processing system, theoretical developments made substantial progress in highlighting thermodynamics limitations and advantages (*Sartori et al., 2014*; *Barato et al., 2014*; *Lan et al., 2012*), making a step towards linking information and dissipation from a molecular perspective (*Ouldridge et al., 2017*; *Flatt et al., 2023*; *Penocchio et al., 2022*).

Here, we consider an archetypal yet minimal model for sensing that is inspired by biological networks (*Lan et al., 2012*; *Tadres et al., 2022*; *Ma et al., 2009*) and encapsulates all these key ingredients, that is negative feedback, storage, and energy dissipation, and study its response to repeated stimuli. Indeed, in the presence of dynamic environments, it is common for a biological system to keep encountering the same stimulus. Under these conditions, a progressive decay in the amplitude of the response is often observed, both at sensory and molecular levels. In general terms, such adaptive behavior is usually named *habituation* and is a common phenomenon recorded in various systems, from biochemical networks (*Rahi et al., 2017*; *Tadres et al., 2022*; *Jalaal et al., 2020*) to populations of neurons (*Malmierca et al., 2014*; *Shew et al., 2015*; *Marquez-Legorreta et al., 2022*; *Fotowat and Engert, 2023*). In particular, habituation characterizes many neuronal circuits along the sensory-motor processing pathways in most living organisms, either invertebrates or vertebrates (*Malmierca et al., 2014*; *Shew et al., 2015*), where inhibitory feedback mechanisms are believed to modulate the stimulus weight (*Lamiré et al., 2022*; *Fotowat and Engert, 2023*; *Barzon et al., 2025*). Most importantly, the first complete characterization of habituating phenomena dates back to 1966 (*Thompson and Spencer, 1966*), when different hallmarks of habituation in vertebrate animals were characterized. Despite its widespread occurrence across remarkably different scales, the connection between habituation in the animal kingdom and brainless molecular systems has only recently attracted considerable attention. A limited number of dynamical models have been proposed to explore the similarities and differences between the manifestations of these two fundamentally distinct phenomena (*Eckert*

*et al., 2024*; *Smart et al., 2024*). However, dynamical characterizations of habituation still lack a clear identification of the functional role of habituation in regulating information flow, optimal processing, and sensitivity calibration (*Benda, 2021*), and in controlling behavior and prediction during complex tasks (*Bueti et al., 2010*; *Sederberg et al., 2018*; *Palmer et al., 2015*).

In this work, we explicitly compute the information shared between readout molecules and external stimulus over time. We find that the information gain peaks at intermediate levels of habituation, uncovering that optimal processing performances are necessarily tangled with maximal activity reduction. This region of optimal information gain can be retrieved by simultaneously minimizing dissipation and maximizing information in the presence of a prolonged stimulation, hinting at an a priori optimality condition for the operations of biological systems. Our results unveil the role of habituation in enhancing processing abilities and open the avenue to understanding the emergence of basic learning mechanisms in simple molecular scenarios.

## Results

### Archetypal model for sensing in biological systems

Several minimal models for adaptation are composed of three building blocks (*Ma et al., 2009*; *Tadres et al., 2022*; *Tu et al., 2008*; *Celani et al., 2011*; *Rahi et al., 2017*): one responsible for buffering the input signal; one representing the output; and one usually reminiscent of an internal memory. Here, we start with an analogous archetypal architecture. The three building blocks (or units) are represented by a receptor ($R$), and readout ($U$) and storage ($S$) populations.

To introduce our model in general terms, we consider a time-varying environment $H$, representing an external signal characterized by a probability $p_H(h, t)$ of being equal to $h$ at time $t$. This input signal is read by the receptor unit $R$. The receptor can be either active ($A$), taking the value $r = 1$, or passive ($P$), $r = 0$, with these two states separated by an energetic barrier $\Delta E$. The transitions between passive and active states can happen through two different pathways, a 'sensing' reaction path (superscript $H$) that is stimulated by the external signal $h$, and an 'internal' path (superscript $I$) that mediates

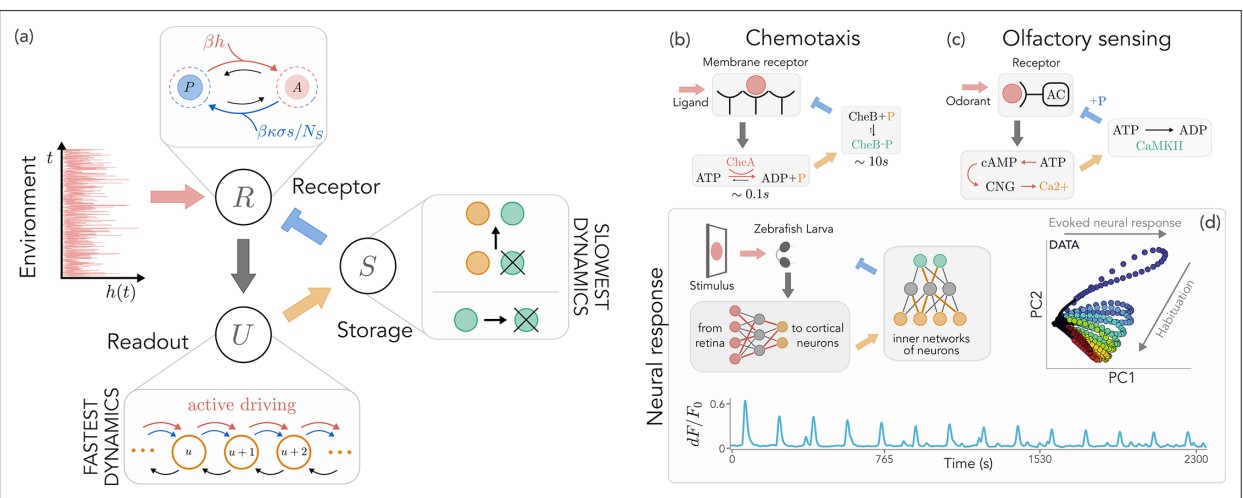

**Figure 1.** Sketch of the model architecture and biological examples at different scales. (**a**) A receptor $R$ transitions between an active ($A$) and passive ($P$) state along two pathways, one used for sensing (red) and affected by the environment $h$, and the other (blue) modified by the energy of storage molecules, $\sigma s$, tuned by inhibition strength $\kappa$ and storage capacity $N_S$. Here, $\beta = (k_B T)^{-1}$ encodes the inverse temperature. An active receptor increases the response of a readout population $U$ (orange), which in turn stimulates the production of storage units $S$ (green) that provide negative feedback to the receptor. (**b**) In the chemical network underlying chemotactic response, we can identify a similar architecture. The input ligand binds to membrane receptors, decreasing kinase activity and producing phosphate groups whose concentration regulates the receptor methylation level. (**c**) Similarly, in olfactory sensing, odorant binding induces the activation of adenylyl cyclase (AC). AC stimulates a calcium flux, eventually producing phosphorylase calmodulin kinase II (CAMKII) which phosphorylates and deactivates AC. (**d**) In neural response, multiple mechanisms take place at different scales. In zebrafish larvae, visual stimulation is projected along the visual stream from the retina to the cortex, a coarse-grained realization of the $R$-$U$ dynamics. Neural habituation emerges upon repeated stimulation, as measured by calcium fluorescence signals ($dF/F_0$) and by the corresponding two-dimensional PCA of the activity profiles.

the effect of the negative feedback from the storage unit (see *Figure 1a*). We further assume, for simplicity, that the rates follow an effective Arrhenius' law:

$$\Gamma_{P \to A}^{(H)} = e^{\beta(h - \Delta E)} \Gamma_R^{(H)} \qquad \Gamma_{A \to P}^{(H)} = \Gamma_R^{(H)}$$
$$\Gamma_{P \to A}^{(I)} = e^{-\beta \Delta E} \Gamma_R^{(I)} \qquad \Gamma_{A \to P}^{(I)} = \Gamma_R^{(I)} e^{\beta \kappa \sigma s / N_S} \tag{1}$$

where the input is modeled as an additional thermodynamic driving with an energy $\beta h$, and $\Gamma_R^{(H)} = g \Gamma_R^{(I)} = \tau_R^{-1}$ sets the timescale of the receptor. In particular, $g$ represents the ratio between the timescales of the two pathways, and the inverse temperature $\beta = (k_B T)^{-1}$ encodes the role of the thermal noise, as lower values of $\beta$ are associated with faster reactions.

The negative feedback depends on the energy provided by the storage, $\sigma s$, where $s$ is the number of active storage molecules. The parameter $\kappa$ represents the strength of the inhibition, and $N_S$ is the storage capacity. For ease of interpretation, we assume that the activation rate of the receptor due to a reference signal $H_{\text{ref}}$ is balanced by the deactivation rate provided by the feedback of a fraction $\alpha = \langle S \rangle / N_S$ of average active storage population:

$$\left\langle \log \frac{\Gamma_{P \to A}^{(H)}}{\Gamma_{A \to P}^{(I)}} \right\rangle = \beta g \left( H_{\text{ref}} - \kappa \sigma \alpha \right) = 0 \quad \to \quad \kappa = \frac{H_{\text{ref}}}{\alpha \sigma} . \tag{2}$$

This condition sets the inhibition strength by choosing the inhibiting fraction $\alpha$. At this stage, the reference signal represents the typical environmental stimulus to which the system is exposed. This choice rationalizes the physical meaning of the model parameters, but it does not alter the phenomenology of the system. Crucially, the presence of two different transition pathways, motivated by molecular considerations and pivotal in many energy-consuming biochemical systems (*De Los Rios and Barducci, 2014*; *Astumian, 2019*; *Flatt et al., 2023*), creates an internal non-equilibrium cycle in receptor dynamics. Without the storage population, the internal pathway would not be present and the receptor would satisfy an effective detailed balance.

Whenever active, the receptor drives the production of readout population $U$, which represents the direct response of the system to environmental signals. As such, it is the observable characterizing habituation (see *Figure 1a*). We model its dynamics with a controlled stochastic birth-and-death process (*Yan et al., 2019*; *Hilfinger et al., 2016*; *Nicoletti and Busiello, 2024a*):

$$\varnothing_U \xrightarrow{\Gamma_{u \to u+1}^{(r)}} U \qquad U \xrightarrow{\Gamma_{u+1 \to u}} \varnothing_U$$
$$\Gamma_{u \to u+1} = e^{-\beta(V - cr)} \Gamma_U^0 \qquad \Gamma_{u+1 \to u} = (u + 1) \Gamma_U^0 \tag{3}$$

where $u$ denotes the number of molecules, $\Gamma_U^0 = \tau_U^{-1}$ sets the timescale of readout production, and $V$ is the energy needed to produce a readout unit. When the receptor is active, $r = 1$, this energetic cost is reduced by an effective additional driving $\beta c$. Active receptors transduce the environmental energy into an active pumping in the readout unit, allowing readout population to encode information on the external signal.

Finally, readout units stimulate the production of the storage population $S$. Its number of molecules $s$ follows again a controlled birth-and-death process:

$$\varnothing_S \xrightarrow{\Gamma_{s \to s+1}(u)} S \qquad S \xrightarrow{\Gamma_{s+1 \to s}} \varnothing_S$$
$$\Gamma_{s \to s+1}(u) = u e^{-\beta \sigma} \Gamma_S^0 \qquad \Gamma_{s+1 \to s} = (s + 1) \Gamma_S^0 \tag{4}$$

where $\sigma$ is the energetic cost of a storage molecule and $\Gamma_S^0$ sets the timescale, i.e., $\Gamma_S^0 = \tau_S^{-1}$. For simplicity, we assume that readout molecules can catalytically activate storage molecules from a passive pool (see *Figure 1a*). Storage units are responsible for encoding the response, playing the role of a finite-time memory.

Our architecture, being devoid of specific biological details, can be adapted to describe systems operating at very different scales (*Figure 1b–d*). However, we emphasize that the proposed model is intentionally oversimplified compared to realistic biochemical or neural systems, yet it contains the minimal ingredients for habituation to emerge naturally. As such, the examples shown in *Figure 1b–d* are meant solely to illustrate the core architecture. In particular, while receptors can be readily identified, the role of readout is played by photo-receptors or calcium concentration for olfactory or visual

sensing mechanisms (*Menini, 1999*; *Kohn, 2007*; *Lesica et al., 2007*; *Benucci et al., 2013*; *Benda, 2021*; *Marquez-Legorreta et al., 2022*; *Fotowat and Engert, 2023*), while storage may represent different molecular mechanisms at a coarse-grained level as, for example, memory molecules sensitive to calcium activity (*Coultrap and Bayer, 2012*), synaptic depotentiation, and neural populations that regulate neuronal response (*Marquez-Legorreta et al., 2022*; *Fotowat and Engert, 2023*).

As a final remark, we expect from previous studies (*Nicoletti and Busiello, 2024a*) that the presence of multiple timescales in the system will be fundamental in shaping information between the different components. Thus, we employ the biologically plausible assumption that $U$ undergoes the fastest evolution, while $S$ and $H$ are the slowest degrees of freedom (*Celani et al., 2011*; *Ngampruetikorn et al., 2020*). We have that $\tau_U \ll \tau_R \ll \tau_S \approx \tau_H$, where $\tau_H$ is the timescale of the environment.

## The hallmarks of habituation

Habituation occurs when, upon repeated presentation of the same stimulus, a progressive decrease to an asymptotic level is observed in some parameters (*Thompson and Spencer, 1966*; *Eckert et al., 2024*). In our model, the response of the system is represented by the average number of active readout units, $\langle U \rangle(t)$. This behavior resembles recent observations on habituation under analogous external conditions in various experimental systems (*Rahi et al., 2017*; *Jalaal et al., 2020*; *Tadres et al., 2022*; *Marquez-Legorreta et al., 2022*; *Fotowat and Engert, 2023*). However, habituation in its strict sense is not sufficient to encompass the diverse array of emergent features recorded in biological systems. In fact, several other hallmarks are closely associated with habituating behavior (*Thompson and Spencer, 1966*; *Smart et al., 2024*; *Eckert et al., 2024*):

1. Potentiation of habituation — After a train of stimulations and a subsequent short pause, the response decrement becomes more rapid and/or more pronounced.
2. Spontaneous recovery — If, after response decrement, the stimulus is suppressed for a sufficiently long time, the response recovers at least partially at subsequent stimulations.
3. Subliminal accumulation — The effect of stimulation may accumulate after the habituation level, thus delaying the onset of spontaneous recovery.
4. Intensity sensitivity — Other conditions being fixed, the less intense the stimulus, the more rapid and/or pronounced the response decrease.
5. Frequency sensitivity — Other conditions being fixed, more frequent stimulation results in a more rapid and/or more pronounced response decrease.

These hallmarks have been originally proposed from observations of vertebrate animals, but they are not the sole properties characterizing the most general definition of habituation. However, the list above encompasses the features that can be obtained from a single stimulation, as in our case, and without any ambiguity in the interpretation (for a detailed discussion, we refer to *Thompson and Spencer, 1966*; *Eckert et al., 2024*).

To explore the ability of the proposed archetypal mode to capture the aforementioned hallmarks, we consider the simple case of an exponential input distribution, $p_H(h, t) \sim \exp\left[-h \langle H \rangle(t)\right]$ with uncorrelated signals, that is $\langle h(t)h(t') \rangle = \langle H \rangle(t) \langle H \rangle(t')$. The time-dependent average $\langle H \rangle$ periodically switches between two values, $\langle H \rangle_{\min}$ and $\langle H \rangle_{\max}$, corresponding to a (non-zero) background signal and a (strong) stimulation of the receptor, respectively. The system dynamics is governed by four different operators, $\hat{W}_X$, with $X = R, U, S, H$, one for each unit and one for the environment. The resulting master equation is:

$$\partial_t P = \left[ \frac{\hat{W}_R(s,h)}{\tau_R} + \frac{\hat{W}_U(r)}{\tau_U} + \frac{\hat{W}_S(u)}{\tau_S} + \frac{\hat{W}_H}{\tau_H} \right] P, \tag{5}$$

where $P$ denotes, in general, the joint propagator $P(u, r, s, h, t|u_0, r_0, s_0, h_0, t_0)$, with $u_0$, $r_0$, $s_0$ and $h_0$ initial conditions at time $t_0$. By taking advantage of the timescale separation, we can write an exact self-consistent solution to *Equation 8* at all times $t$ (see Materials and methods and Supplementary Information).

In *Figure 2a*, we show that the system exhibits habituation in its strict sense. Here, for simplicity, we consider a train of signals arriving at times $t_1, \ldots, t_N$, each lasting a time $T_s$ with equal pauses between them of duration $\Delta T$. We define the time to habituate, $t^{(\text{hab})}$, as the first time at which the relative change of our observable, $\langle H \rangle(t)$, is less than 0.5%, in analogy to *Eckert et al., 2024*. Clearly, $t^{(\text{hab})}$ is associated with a number of stimuli necessary to habituate, $n^{(\text{hab})}$, i.e.,

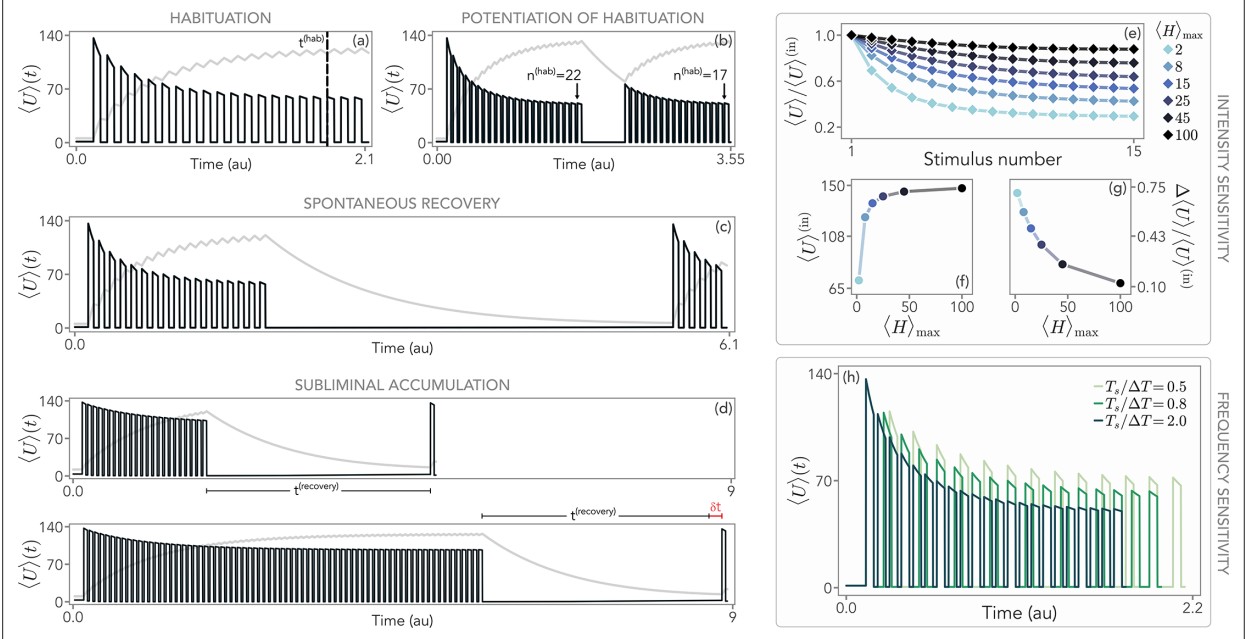

**Figure 2.** Hallmarks of habituation. (**a**) An external signal switch between two values, $\langle H \rangle_{\min} = 0.1$ (background) and $\langle H \rangle_{\max} = H_{\mathrm{ref}} = 10$ (stimulus). The inter-stimuli interval is $\Delta T = 100 \,(\text{a.u.})$ and the duration of each stimulus $T_s = 100 \,(\text{a.u.})$. The average readout population (black) follows the stimulation, increasing when the stimulus is presented. The response decreases upon repeated stimulation, signaling the presence of habituation. Conversely, the average storage population (gray) increases over time. The black dashed line represents the time to habituate $t^{(\mathrm{hab})}$ (**Equation 9**). (**b**) If the stimulus is paused and presented again after a short time, the system habituates more rapidly, that is the number of stimulations to habituate $n^{(\mathrm{hab})}$ is reduced. (**c**) After waiting a sufficiently long time, the response can be fully recovered. (**d**) If the stimulation continues beyond habituation, the time to recover the response $t^{(\mathrm{recovery})}$ (**Equation 10**) increases by an amount $\delta t$ (in red). (**e**) The relative decrement of the average readout with respect to the initial response, $\langle U \rangle^{(\mathrm{in})}$, shows that habituation becomes less and less pronounced as we increase $\langle H \rangle_{\max}$. (**f**) As expected, the initial response increases with $\langle H \rangle_{\max}$. (**g**) The relative difference between $\langle H \rangle \, (t^{(\mathrm{hab})})$ and $\langle U \rangle^{(\mathrm{in})}$, $\Delta \langle U \rangle$, decreases with the stimulus strength. (**h**) By changing $\Delta T$ and keeping the stimulus duration $T_s$ fixed, we observe that more pronounced and more rapid response decrements are associated with more frequent stimulation. Parameters are reported in the Methods, and these hallmarks are qualitatively independent of their specific choice.

$$\frac{\langle U \rangle \, (t_{n^{(\mathrm{hab})}-1}) - \langle U \rangle \, (t_{n^{(\mathrm{hab})}} \equiv t^{(\mathrm{hab})})}{\langle U \rangle \, (t_{n^{(\mathrm{hab})}})} \leq 0.005 \tag{6}$$

Our results do not qualitatively change when choosing a different threshold. Hallmark 1, potentiation of habituation, corresponds to a reduction of $n^{(\mathrm{hab})}$ after one series of stimulation and recovery. This implies a more rapid decrement in the response and a shorter time to achieve habituation, as we show in **Figure 2b**. Analogously, hallmark 2 is presented in **Figure 2c**, where we show that by suppressing the stimulus for a sufficiently long amount of time, the response spontaneously recovers to the pre-habituation level. Furthermore, by stimulating the system beyond $t^{(\mathrm{hab})}$, we also observe an increase in the amount of time to achieve complete recovery (hallmark 3). We define this recovery period $t^{(\mathrm{recovery})}$ as the first time required to have a response with a relative strength not greater than 1% with respect to the one at the first stimulus, that is

$$\frac{\langle U \rangle \, (t_1) - \langle U \rangle \, (t^{(\mathrm{recovery})})}{\langle U \rangle \, (t_1)} \leq 0.01. \tag{7}$$

In **Figure 2d**, we show that the recovery period increases by $\sim 5\%$ as a consequence of this subliminal accumulation.

Within the same setting, in **Figure 2e–g** we applied stimuli of different strengths $\langle H \rangle_{\max}$ to study the sensitivity to input intensity (hallmark 4). When normalized by the initial response $\langle U \rangle^{(\mathrm{in})} \equiv \langle U \rangle \, (t_1)$, less intense stimuli result in stronger response decrements (see **Figure 2e**). At the same time, as expected, the absolute value of the initial response increases instead (see **Figure 2f**). Hallmark 4 is clearly captured by **Figure 2g**, where we quantify the decrease of the normalized total habituation level,

$\Delta \langle U \rangle = \langle U \rangle (t^{(\text{hab})}) - \langle U \rangle^{(\text{in})}$, when exposed to increasing $\langle H \rangle_{\max}$. The last feature (hallmark 5) is reported in **Figure 2h**, where we keep the duration of the stimulus $T_s$ fixed while changing the inter-stimuli interval $\Delta T$. By showing the responses up to the habituation time, we clearly notice that more frequent stimulation is associated with a more rapid and more pronounced response decrement.

Summarizing, despite its simplicity and lack of biological details, our model encompasses the minimal ingredients to capture the main hallmarks defining habituation.

## Information from habituation

In our architecture, habituation emerges due to the increase in the storage population, which provides increasing negative feedback to the receptor and thus lowers the number of active readout units $\langle U \rangle (t)$. Crucially, by solving the master equation in **Equation 8**, we can also study the evolution of the full probability distribution $p_{U,S,H}(t)$. This approach allows us to quantify how the system encodes information on the environment $H$ through its readout population and how it changes during habituation. To this end, we introduce the mutual information between $U$ and $H$ at time $t$ (see Materials and methods):

$$I_{U,H}(t) = \mathcal{H}[p_U](t) - \int_0^\infty dh \, p_H(h,t) \mathcal{H}[p_{U|H}](t) \tag{8}$$

where $\mathcal{H}[p](t)$ is the Shannon entropy of the probability distribution $p$, and $p_{U|H}$ denotes the conditional probability distribution of $U$ given $H$ measures information in terms of statistical dependencies, that is of how factorizable the joint probability distribution $p_{U,H}$ is. It vanishes if and only if $U$ and $H$ are independent. Notably, the mutual information coincides with the entropy increase of the readout distribution:

$$k_B I_{U,H} = -k_B \left( \mathcal{H}[p_{U|H}] - \mathcal{H}[p_U] \right) = -\Delta \mathbb{S}_U \tag{9}$$

where $\Delta \mathbb{S}_U$ is the change in entropy of the readout population due to repeated measurements of the signal (**Parrondo et al., 2015**).

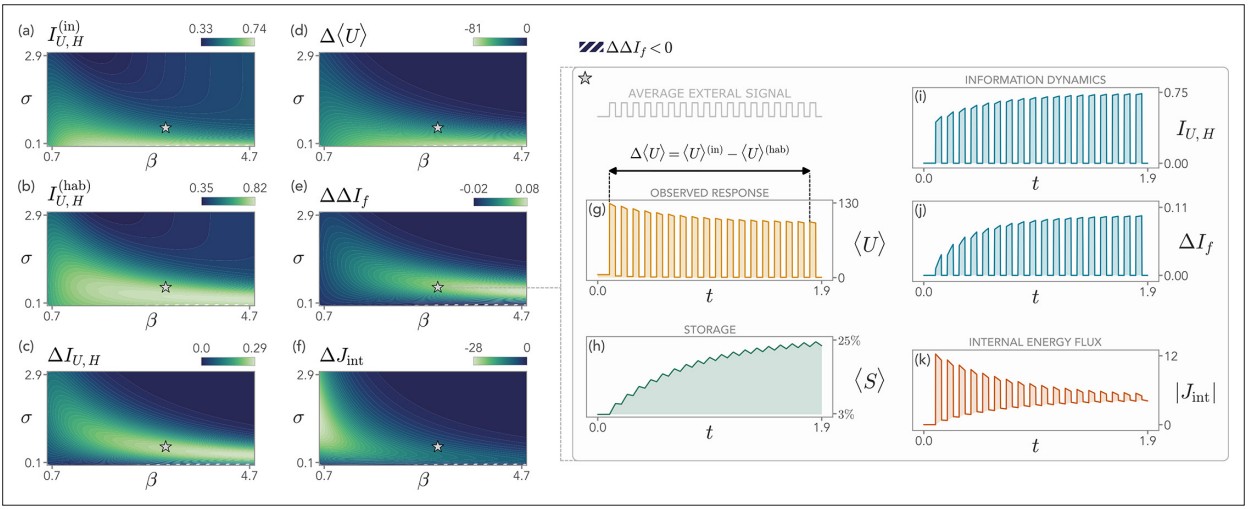

**Figure 3.** Information and thermodynamics of the model during repeated external stimulation, as a function of the inverse temperature $\beta$ and the energetic cost of storage $\sigma$. (**a–b**) The mutual information between readout population and external signal at the first stimulus, $I_{U,H}^{(\text{in})}$, is typically lower than the one when the system has habituated, $I_{U,H}^{(\text{hab})}$. (**c**) The change in the mutual information, $\Delta I_{U,H}$, displays a peak in a region of the $(\beta, \sigma)$ space, where the system exhibits optimal information gain during habituation. (**d**) This region corresponds to intermediate habituation strength, as measured by $\Delta \langle U \rangle$. (**e**) The corresponding increase in the feedback information $\Delta I_f$ indicates that storage is fostering the gain in $\Delta I_{U,H}$. (**f**) Habituation promotes a decrease of the internal energy flux $\Delta J_{\text{int}}$, suggesting a synergistic energetic advantage of habituation. (**g–h**) From the dynamical point of view, in the region of maximal information gain ($\beta = 3$, $\sigma = 0.6$) the average number of readout units, $\langle U \rangle$, decreases over time, while the average storage population, $\langle S \rangle$, increases. (**i–j**) Similarly, both the information encoded on $H$ by the readout, $I_{U,H}$, and the feedback information, $\Delta I_f$, increase upon repeated stimulations. (**k**) The absolute value of the internal energy flux, $|J_{\text{int}}|$, decreases upon stimulations, while increasing for repeated pauses when the system moves downhill in energy. Model parameters are as specified in the Methods, $\langle H \rangle_{\min} = 0.1$, and $\langle H \rangle_{\max} = H_{\text{ref}} = 10$.

As in the previous section, we considered a switching signal with $\langle H \rangle_{\text{max}} = H_{\text{ref}}$, the typical environmental stimulus strength. In **Figure 3a–b**, we plot the mutual information at the first signal, $I_{U,H}^{(\text{in})}$, and when the system has habituated, $I_{U,H}^{(\text{hab})}$, as a function of $\beta$ and $\sigma$. Crucially, we find that there exist parameters for which $I_{U,H}^{(\text{hab})}$ is larger than $I_{U,H}^{(\text{in})}$. This result suggests that the information on $H$ encoded by $U$ in the habituated system is larger than the initial one. We can quantify this effect by introducing the mutual information gain

$$\Delta I_{U,H} = I_{U,H}^{(\text{hab})} - I_{U,H}^{(\text{in})}. \tag{10}$$

In **Figure 3c**, we show that $\Delta I_{U,H}$ displays a peak in an intermediate region of the $(\beta, \sigma)$ plane. In this region, the corresponding habituation strength

$$\Delta \langle U \rangle = \langle U \rangle^{(\text{hab})} - \langle U \rangle^{(\text{in})} \tag{11}$$

attains intermediate values, suggesting that too strong habituation can be detrimental (**Figure 3d**). This behavior is tightly related to the presence of the storage $S$, which acts as an information reservoir for the system. To rationalize this feature, we introduce the feedback information

$$\Delta I_f = I_{(U,S),H} - I_{U,H} > 0 \tag{12}$$

quantifying how much the simultaneous knowledge of $U$ and $S$ increases information compared to $U$ alone. Indeed, the change in feedback information after habituation, $\Delta\Delta I_f = \Delta I_f^{(\text{hab})} - \Delta I_f^{(\text{in})}$, peaks in the same region of $\Delta I_{U,H}$ (**Figure 3e**).

For small $\sigma$ we find that $\Delta\Delta I_f$ may become negative, indicating that a too strong storage production may ultimately impede the information-theoretic performances of the system. Moreover, producing storage molecules requires energy. We can compute the internal energy flux associated with the storage of information through $S$ as

$$J_{\text{int}} = \sigma \sum_{u,s} \left[ \Gamma_{s \to s+1} p_{U,S^{(u,s,t)}} + \right.$$
$$\left. - \Gamma_{s+1 \to s} p_{U,S^{(u,s+1,t)}} \right], \tag{13}$$

which is the total energy flux to produce the internal populations ($U$ and $S$), since $U$ always reaches equilibrium, being the fastest species at play. Its change during habituation is defined as $\Delta J_{\text{int}} = J_{\text{int}}^{(\text{hab})} - J_{\text{int}}^{(\text{in})}$. In **Figure 3f**, we show that $\Delta J_{\text{int}}$ is typically smaller than zero, hinting at a synergistic thermodynamic advantage of habituation.

In **Figure 3g–k**, we show the evolution of the system for values of $(\beta, \sigma)$ that lie in the region of maximal information gain. The readout activity decreases in time (**Figure 3g**), due to the habituation driven by the increase of $\langle S \rangle$ (**Figure 3h**). In this region, both $I_{U,H}$ and $\Delta I_f$ increase over time (**Figure 3i–j**). We note that the increase in $I_{U,H}$ is concomitant to a reduction of the population that is encoding the signal. Although this may seem surprising, we stress that the mean of $U$ is not directly related to the factorizability of the joint distribution $p_{U,H}$. Finally, in **Figure 3k**, we show that the absolute value of the internal energy flux |$J_{\text{int}}$| in the presence of the stimulus sharply decreases as well, while increasing during its pauses (the value of $J_{\text{int}}$ is negative in the presence of the background signal since the system is moving downhill in energy). This behavior is due to the interplay between storage and readout populations during habituation and signals the fact that the system requires progressively less energy to respond as time passes, while also moving less downhill in energy when the stimulus is paused. This observation suggests that the regime of maximal information gain supports habituation with a concurrent energetic advantage.

## The onset of habituation and its functional role

As habituation, information, and their energetic cost appear to be tightly related, we now investigate whether the region of maximal information gain can be retrieved by means of an a priori optimization principle. To do so, we first focus on the case of a constant environment. We assume that the system can tune its internal parameters to optimally respond to the statistics of a prolonged external signal.

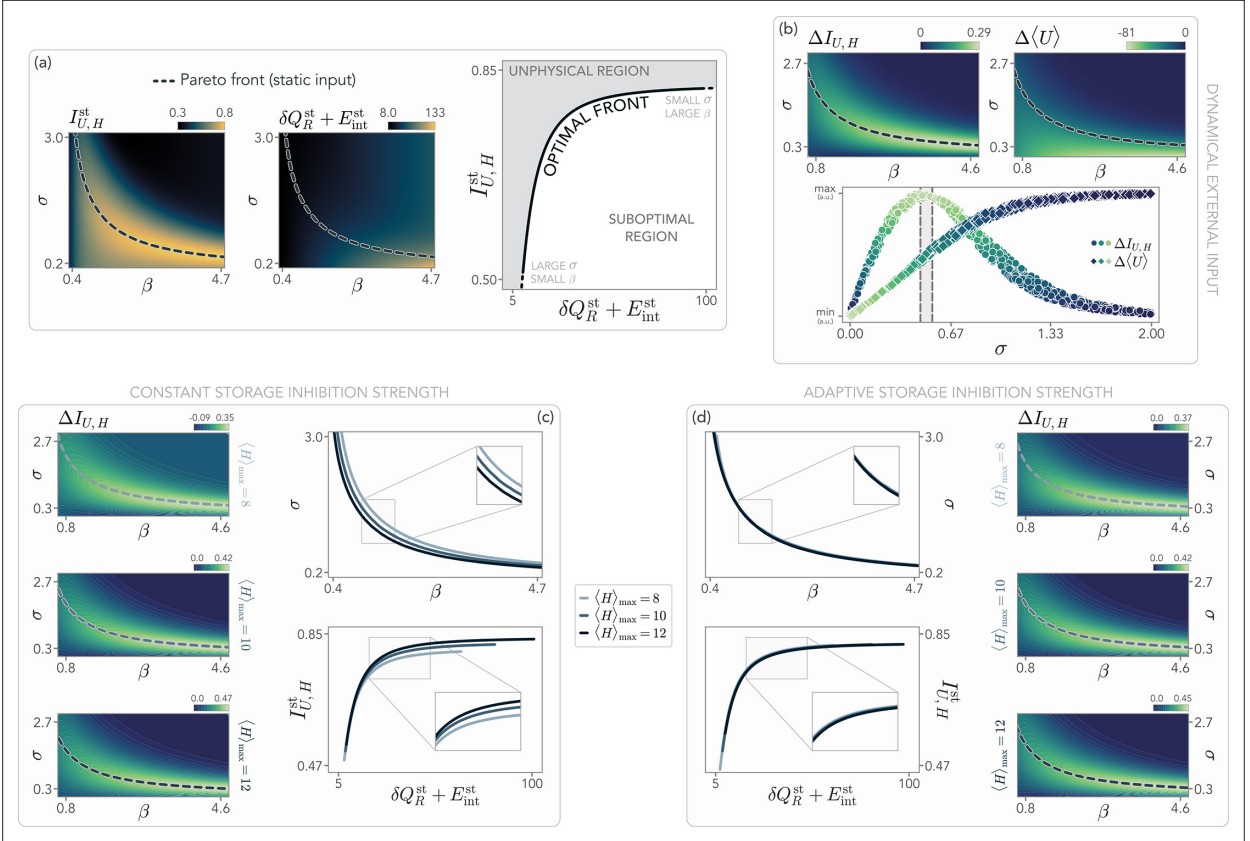

**Figure 4.** Optimality at the onset of habituation and dependence on the external signal strength. (**a–b**) Contour plots in the $(\beta, \sigma)$ plane of the stationary mutual information $I^{st}_{U,H}$, and the total dissipation of the system per unit energy, $\delta Q^{st}_R + E^{st}_{int}$, in the presence of a constant signal $\langle H \rangle = H_{ref} = 10$. For a given value of $\beta$, the system can optimize $\sigma$ to the Pareto front (black line) to simultaneously minimize energy consumption and maximize information. Below the front, the system exploits the available energy suboptimally, while the region above the front is physically inaccessible. (**b**) In the presence of a dynamical input switching between $\langle H \rangle_{min} = 0.1$ and $\langle H \rangle_{max} = H_{ref}$, the parameters defining the optimal front capture the region of maximal information gain corresponding to the onset of habituation, where $\Delta \langle U \rangle$ starts to be significantly smaller than zero. The gray area enclosed by the dashed vertical lines indicates the location of the Pareto front for values of $\beta \in [3 - 3.5]$. (**c**) The Pareto front depends on the strength of the external signal $\langle H \rangle_{max}$. In particular, for $\langle H \rangle_{max} < H_{ref}$, at fixed $\beta$ a larger storage cost $\sigma$ is needed. Conversely, for $\langle H \rangle_{max} > H_{ref}$, an optimal system can harvest more information by producing more storage, thus exhibiting a smaller $\sigma$. (**d**) If we allow the system to adapt its inhibition strength $\kappa$ to the stimulus (**Equation 16**), the Pareto fronts for different external signals collapse into a single optimal curve. Model parameters are specified in the Materials and methods.

Thus, we consider a fixed input statistics given by $p^{st}_H(h) \sim \exp[-h/H^{st}]$, with $H^{st}$ the average signal strength.

When the system reaches its steady state, we compute the information that the readout has on the signal, $I^{st}_{U,H}$ (**Figure 4a**) and the total energy consumption. To this end, we must take into account two terms. First, the energy flux in **Equation 13** represents the rate of change in energy due to the driven storage production. The energy consumption associated with this process per unit energy is $E^{st}_{int} = \tau_S J^{st}_{int}/\sigma$. Second, the inhibition pathway is also driving the receptor out of equilibrium, leading to a dissipation per unit temperature given by

$$\delta Q_R = \left\langle \log \left( \frac{\Gamma^{(H)}_{P \to A} \Gamma^{(I)}_{A \to P}}{\Gamma^{(H)}_{A \to P} \Gamma^{(I)}_{P \to A}} \right) \right\rangle = \beta \left( H^{st} + \kappa \sigma \frac{\langle S \rangle}{N_S} \right) . \tag{14}$$

We plot the total energy consumption per unit energy $E^{st}_{tot} = \delta Q^{st}_R + E^{st}_{int}$ in **Figure 4a**. In order to understand how the system may achieve large values of mutual information while minimizing its intrinsic dissipation, we can maximize the Pareto functional (**Seoane and Solé, 2015**; **Nicoletti and Busiello, 2024b**):

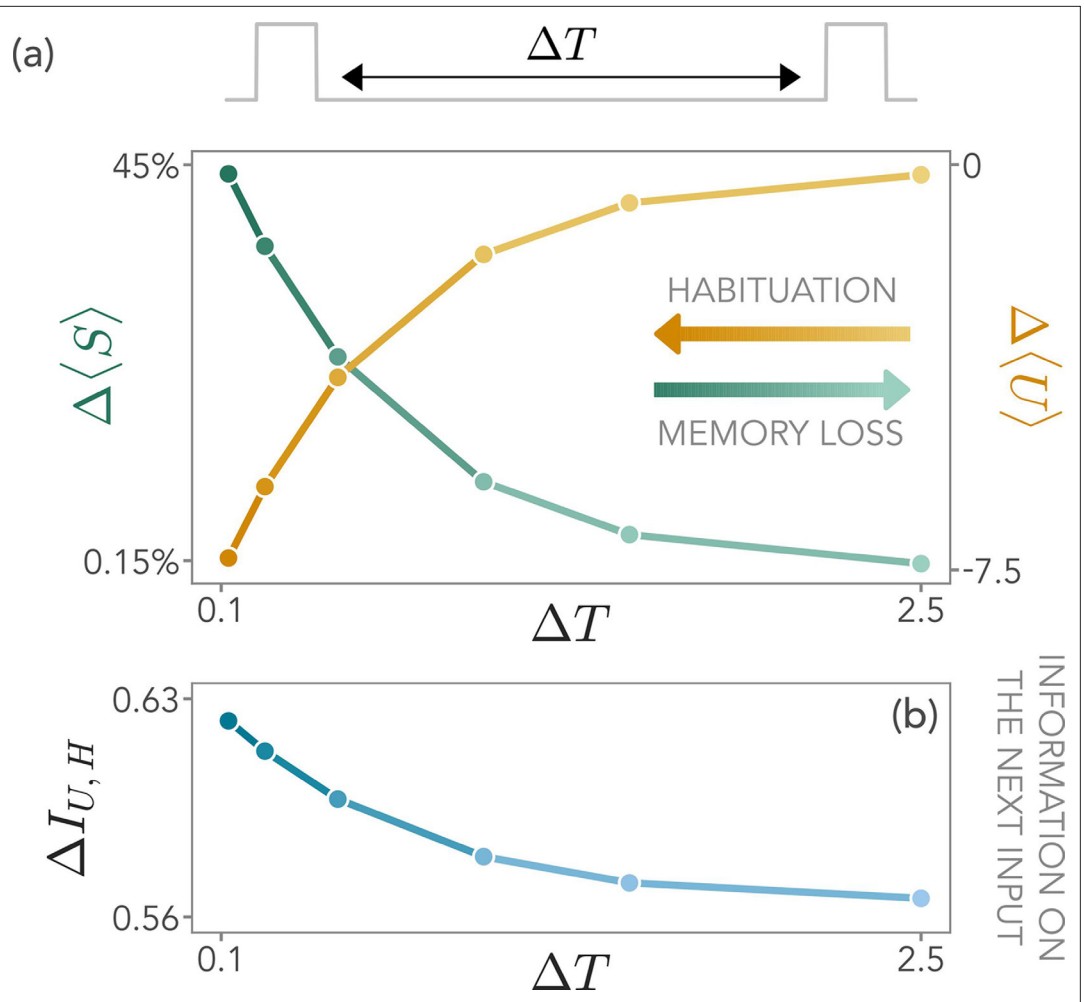

**Figure 5.** The role of memory in shaping habituation. (**a**) The system response depends on the waiting time $\Delta T$ between two external signals. As $\Delta T$ increases, the storage decays, and thus memory is lost (green). Consequently, the habituation of the readout population decreases (yellow). (**b**) As a consequence, the information $I_{U,H}$ that the system has on the signal $H$ when the new stimulus arrives decays as well. Model parameters for this figure are $\beta = 2.5$, $\sigma = 0.5$ in the unit measure of the energy, and as specified in the Materials and methods.

$$\mathcal{L}(\beta, \sigma) = \gamma I_{U,H}^{\mathrm{st}}(\beta, \sigma) - (1 - \gamma) E_{\mathrm{tot}}^{\mathrm{st}}(\beta, \sigma) \tag{15}$$

where $\gamma \in [0, 1]$ sets the strategy implemented by the system. If $\gamma \ll 1$, the system prioritizes minimizing dissipation, whereas if $\gamma \approx 1$ it acts to preferentially maximize information. The set of $(\beta, \sigma)$ that maximize *Equation 15* defines a Pareto optimal front in the $(E_{\mathrm{tot}}^{\mathrm{st}}, I_{U,H}^{\mathrm{st}})$ space (*Figure 4a*). At fixed energy consumption, this front represents the maximum information between the readout and the external input that can be achieved. The region below the front is therefore suboptimal. Instead, the points above the front are inaccessible, as higher values of $I_{U,H}^{\mathrm{st}}$ cannot be attained without increasing $E_{\mathrm{tot}}^{\mathrm{st}}$. We note that, since $\beta$ usually cannot be directly controlled by the system, the Pareto front indicates the optimal $\sigma$ to which the system tunes at fixed $\beta$ (see Materials and methods and Appendices for details).

We now consider once more a system receiving a dynamically switching signal with $\langle H \rangle_{\mathrm{max}} = H^{\mathrm{st}}$. We first focus on the case $H_{\mathrm{ref}} = H^{\mathrm{st}}$, with $H_{\mathrm{ref}}$ the reference signal appearing in *Equation 2*. Remarkably, we find that the Pareto optimal front in the $(\beta, \sigma)$ plane qualitatively corresponds to the region of maximal information gain, as we show in *Figure 4b*. This implies that a system that has tuned its internal parameters to respond to a constant signal also learns how to respond optimally to the time-varying input of the same strength, in terms of information gain. Since the region identified by the front leads to intermediate values of $\Delta \langle U \rangle$, it corresponds to the 'onset of habituation', where

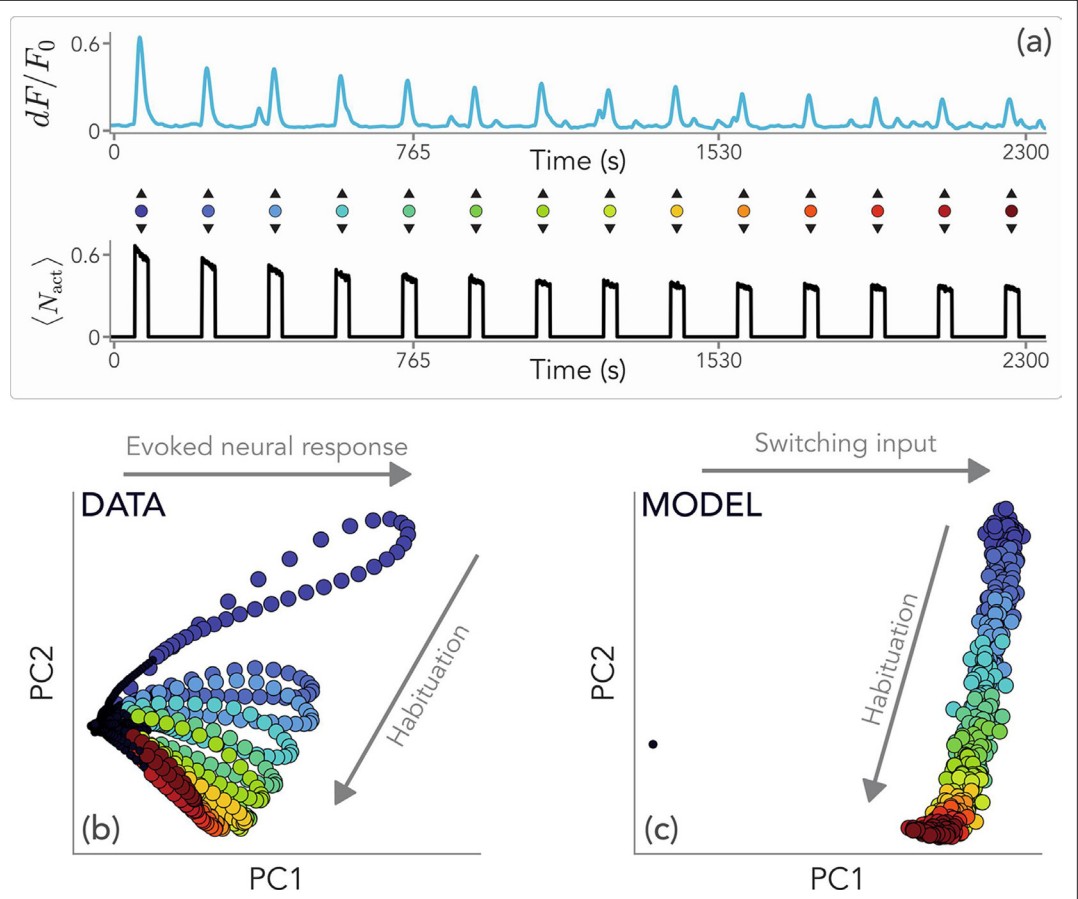

**Figure 6.** Habituation in zebrafish larvae. (**a**) Normalized neural activity profile in a zebrafish larva in response to the repeated presentation of visual (looming) stimulation, and comparison with the fraction of active neurons $\langle N \rangle_{\mathrm{act}} = N_{\mathrm{act}}/N$ in our model with stochastic neural activation (see Methods). Stimuli are indicated with colored dots from blue to red as time increases. (**b**) PCA of experimental data reveals that habituation is captured mostly by the second principal component, while features of the evoked neural response are captured by the first one. Different colors indicate responses to different stimuli. (**c**) PCA of simulated neural activations. Although we cannot capture the dynamics of the evoked neural response with a switching input, the core features of habituation are correctly captured along the second principal component. Model parameters are $\beta = 4.5$, $\sigma = 0.15$ in energy units, and as in the Materials and methods, so that the system is tuned to the onset of habituation.

the system decreases its response enough to reduce the energy dissipation while storing information to increase $I_{U,H}$. Heuristically, the onset of habituation emerges spontaneously when the system attempts to activate its receptor as little as possible, while producing the minimum amount of storage molecules retaining enough information about the external environment.

In *Figure 4c*, we then study what happens to the optimal front if $\langle H \rangle_{\mathrm{max}}$ is larger or smaller than the reference signal. We find that, at low $\langle H \rangle_{\mathrm{max}}$, the Pareto front moves in such a way that a larger storage cost $\sigma$ is needed at fixed $\beta$. This is expected since, at lower signal strengths, it is harder for the system to distinguish the input from the background thermal noise. Conversely, when $\langle H \rangle_{\mathrm{max}} > H_{\mathrm{ref}}$, an optimal system, it needs to reduce $\sigma$ to produce more storage and harvest information. Importantly, we find that if $\langle H \rangle_{\mathrm{max}}$ remains close to $H_{\mathrm{ref}}$, the optimal front remains close to the onset of habituation and thus lies within the region of maximal information gain.

However, we can achieve a collapse of the optimal front if we allow the system to tune the inhibition strength $\kappa$ to the value of the external signal, that is

$$\kappa(\langle H \rangle_{max}) = \frac{\langle H \rangle_{\mathrm{max}}}{\alpha \, \sigma} \; . \tag{16}$$

In this way, a stronger input will correspond to a larger $\kappa$, and thus a stronger inhibition. In *Figure 4d*, we show that the Pareto fronts obtained with this choice collapse into a single curve. Crucially, this front still corresponds to the region of maximal information gain, although the specific values of $\Delta I_{U,H}$ naturally depend on $\langle H \rangle_{\max}$ (see Supplementary Information). Thus, in this scenario, a system that is capable of adapting the negative feedback to its environment is also able to always tune itself to the onset of habituation at different values of the external stimulus and without tinkering with the energy cost $\sigma$, where its responses are optimal from an information-theoretic perspective.

## The role of information storage

The presence of a storage mechanism is fundamental in our model. Furthermore, its role in mediating the negative feedback is suggested by several experimental and theoretical observations (*Celani et al., 2011*; *Tu et al., 2008*; *Kollmann et al., 2005*; *Barkai and Leibler, 1997*; *De Ronde et al., 2010*; *Selimkhanov et al., 2014*). Whenever the storage is eliminated from our model, habituation cannot take place, highlighting its key role in driving the observed dynamics (see Supplementary Information).

In *Figure 5a*, we show that the degree of habituation, $\Delta \langle U \rangle$, and the change in the storage population, $\Delta \langle S \rangle$, are deeply related to one another. The more $\langle S \rangle$ relaxes between two consecutive signals, the less the readout population reduces its activity. This ascribes to the storage population the role of an effective memory and highlights its dynamical importance for habituation. Moreover, the dependence of the storage dynamics on the interval between consecutive signals, $\Delta T$, influences information gain as well. Indeed, increasing $\Delta T$, we observe a decrease of the mutual information (*Figure 5b*) on the next stimulus. In the Supplementary Information, we further analyze the impact of different signal and pause durations.

We remark here that the proposed model is fully Markovian in its microscopic components, and the memory that governs readout habituation spontaneously emerges from the interplay among the internal timescales. In particular, recent works have highlighted that the storage needs to evolve on a slower timescale, comparable to that of the external input, in order to generate information in the receptor and in the readout (*Nicoletti and Busiello, 2024a*). To strengthen our conclusions, we remark that an instantaneous negative feedback implemented directly by $U$ (bypassing the storage mechanism) would lead to no time-dependent modulations of the readout and thus no habituation (see Supplementary Information). Similarly, a readout population evolving on a timescale comparable to that of the signal cannot effectively mediate the negative feedback on the receptor since its population increase would not lead to habituation (see Supplementary Information). Thus, negative feedback has to be implemented by a separate degree of freedom evolving on a timescale which is slow and comparable to that of external signal.

## Minimal features of neural habituation

In neural systems, habituation is typically measured as a progressive reduction of the stimulus-driven neuronal firing rate (*Malmierca et al., 2014*; *Shew et al., 2015*; *Benda, 2021*; *Marquez-Legorreta et al., 2022*; *Fotowat and Engert, 2023*). To test whether our minimal model can be used to capture the typical neural habituation dynamics, we measured the response of zebrafish larvae to repeated looming stimulations via volumetric multiphoton imaging (*Bruzzone et al., 2021*). From a whole-brain recording of $\approx 55000$ neurons, we extracted a subpopulation of $\approx 2400$ neurons in the optic tectum with a temporal activity profile that is most correlated with the stimulation protocol (see Materials and methods).

Our model can be extended to qualitatively reproduce some features of the progressive decrease in neuronal response amplitudes. We identify a single readout unit with a subpopulation of binary neurons. Then, a fraction of neurons is randomly turned on each time the corresponding readout unit is activated (see Materials and methods). We tune the model parameters to have a comparable number of total active neurons at the first stimulus with respect to the experimental setting. Moreover, we set the pause and signal durations in line with the typical timescales of the looming stimulation. We choose the model parameters $\beta$ and $\sigma$ in such a way that the system operates close to the peak of information gain, with an activity decrease over time that is comparable to the activity decrease in experimental data (see Supplementary Information). In this way, we can focus on the effects of storage and feedback mechanisms without modeling further biological details.

The patterns of the model-generated activity are remarkably similar to the experimental ones (see *Figure 6a*). We performed a two-dimensional embedding of the neural activity profiles of all recorded neurons via PCA (explained variance $\approx$ 70%) and we plot the temporal evolution in this low-dimensional space (*Figure 6b*). This procedure reveals that the first principal component (PC) accounts for the evoked neural response, while the second PC mostly reflects the habituation dynamics. We perform the same analysis on data generated from the model as explained above. As we see in *Figure 6c*, the second PC encodes habituation, as in experimental data, although the neural response in the first PC is replaced by the switching on/off dynamics of the input. This shows that our model is able to capture the main features of the observed neural habituation, without the need for biological details.

## Discussion

In this work, we studied a minimal architecture that serves as a microscopic and archetypal description of sensing processes across biological scales. Informed by theoretical and experimental observations, we focused on three fundamental mechanisms: a receptor, a readout population, and a storage mechanism that drives negative feedback. Despite its simplicity, we have shown that our model robustly reproduces the hallmarks associated with habituation in the presence of a single type of repeated stimulation, a widespread phenomenon in both biochemical and neural systems. By quantifying the mutual information between the external signal and readout population, we identified a regime of optimal information gain during habituation. Remarkably, the system can spontaneously tune to this region of parameters if it enforces an information-dissipation trade-off. In particular, optimal systems lie at the onset of habituation, characterized by intermediate levels of activity reduction, as both too-strong and too-weak negative feedback are detrimental to information gain. Finally, we found that, by allowing for a storage inhibition strength that can adapt to the environmental signal, this optimality is input-independent and requires no further adjustment of other internal model parameters. Our results suggest that the functional advantages of the onset of habituation are rooted in the interplay between energy dissipation and information gain, and its general features are tightly linked to the internal mechanisms to store information.

Although minimal, our model can capture basic features of neural habituation, where it is generally accepted that inhibitory feedback mechanisms modulate the stimulus weight (*Lamiré et al., 2022*). Remarkably, recent works reported the existence of a separate inhibitory neuronal population whose activity increases during habituation (*Fotowat and Engert, 2023*). Our model suggests that this population might play the role of a storage mechanism, allowing the system to habituate to repeated signals. However, in neural systems, a prominent role in encoding both short- and long-term information is also played by synaptic plasticity (*Abbott and Nelson, 2000*; *Martin et al., 2000*) as well as by memory molecules (*Coultrap and Bayer, 2012*; *Frankland and Josselyn, 2016*; *Lisman et al., 2002*), at a biochemical level. A comprehensive analysis of how information is encoded and retrieved will most likely require all these mechanisms at once. Including an explicit connectivity structure with synaptic updates in our model may help in this direction, at the price of analytical tractability. Furthermore, future works may be able to compare our theoretical predictions with experiments in which the modulation of frequency (*Fotowat and Engert, 2023*) and intensity of stimulation trigger the observed hallmarks. In this way, we could elucidate the roles and features of internal processes characterizing the system under investigation, along with its information-theoretic performance. Overall, the present results hint at the fact that our minimal architecture may provide crucial insights into the functional advantages of habituation in a wide range of biological systems.

Extensions of these ideas are manifold. The definition of a habituated system relies, in this work as well as in other studies (*Eckert et al., 2024*), on the definition of a response threshold. However, some of the hallmarks might disappear when habituation is defined as a phenomenon appearing in a time-periodic steady state. To overcome this issue, it may be necessary to extend the model to more realistic molecular schemes encompassing the presence of additional storage mechanisms. More generally, understanding the information-theoretic performance of real-world biochemical networks exhibiting habituation remains a fascinating perspective to explore. Upon these premises, the possibility of inferring the underlying biochemical structure from observed behaviors is a fascinating direction (*Rahi et al., 2017*). Furthermore, since we focused on repetitions of statistically identical signals, it will be fundamental to characterize the system's response to diverse environments (*Hidalgo et al., 2014*). To this end, incorporating multiple receptors or storage populations may

be needed to harvest information in complex conditions. In such scenarios, correlations between external signals may help reduce the encoding effort as, intuitively, $S$ is acting as an information reservoir for the system. Moreover, such stored information could be used to make predictions on future stimuli and behavior (*Bueti et al., 2010*; *Sederberg et al., 2018*; *Palmer et al., 2015*). Indeed, living systems do not passively read external signals but often act upon the environment. We believe that both storage mechanisms and their associated negative feedback will remain core modeling ingredients.

Our work paves the way to understanding how information is encoded and guides learning, predictions, and decision-making, a paramount question in many fields. On the one hand, it encapsulates key ingredients to support habituation while still being minimal enough to allow for analytical treatment. On the other hand, it may help the experimental quest for signatures of these physical ingredients in a variety of systems. Ultimately, our results show how habituation – a ubiquitous phenomenon taking place at strikingly different biological scales – may stem from an information-based advantage, shedding light on the optimization principle underlying its emergence and relevance for any biological system.

# Materials and methods
## Model parameters
In this section, we briefly recall the free parameters of the model and the values we use in numerical simulations, unless otherwise specified. In particular, the energetic barrier $(V - cr)$ fixes the average values of the readout population both in the passive and active state, namely $\langle U \rangle_P = e^{-\beta V}$ and $\langle U \rangle_A = e^{-\beta(V-c)}$ (see *Equation 3*). Thus, we can fix $\langle U \rangle_P$ and $\langle U \rangle_A$ in lieu of $V$ and $c$. Similarly, as in *Equation 2*, we can set the inhibiting storage fraction $\alpha$ to fix $\kappa$. At any rate, we remark that the emerging features of the model are qualitatively independent of the specific choice of these parameters. Furthermore, we typically consider the average of the exponentially distributed signal to be $\langle H \rangle_{\max} = 10$ and $\langle H \rangle_{\min} = 0.1$ (see Supplementary Information for details). Overall, we are left with $\beta$ and $\sigma$ as free parameters. $\beta$ quantifies the amount of thermal noise in the system, and at small $\beta$ the thermal activation of the receptor hinders the effect of the signal and makes the system almost unable to process information. Conversely, if $\beta$ is high, the system must overcome large thermal inertia, increasing the dissipative cost. In this regime of weak thermal noise, we expect that, given a sufficient amount of energy, the system can effectively process information. In *Table 1*, we summarize the specific parameter values we used throughout the main text. Other values to explore the robustness of the model are discussed in the Supplementary Information.

**Table 1.** Summary of the model parameters and the values used for numerical simulations, unless otherwise specified.
The parameters $\beta$ and $\sigma$ qualitatively determine the behavior of the model and are varied throughout the main text.

| Parameter | Description | Value |
|---|---|---|
| $M_S$ | Maximum number of storage units | 30 |
| $\Delta E$ | Receptor energetic barrier | 1 |
| $\langle U \rangle_P$ | Average readout with passive receptor | 150 |
| $\langle U \rangle_A$ | Average readout with active receptor | $M_S$ |
| $\Gamma_S^0$ | Inverse timescale of the storage | 1 |
| $g$ | Receptor's pathways timescale ratio | 1 |
| $\alpha$ | Inhibiting storage fraction | 2/3 |
| $H_{\mathrm{ref}}$ | Reference signal | 10 |
| $\beta$ | Inverse temperature | - |
| $\sigma$ | Storage energy cost | - |

## Timescale separation

We solve our system in a timescale separation framework (*Busiello et al., 2020*; *Bo and Celani, 2017*; *Nicoletti and Busiello, 2024a*), where the storage evolves on a timescale that is much slower than all the other internal ones, that is

$$\tau_U \ll \tau_R \ll \tau_S \approx \tau_H .$$

The fact that $\tau_S$ is the slowest timescale at play is crucial to making these components act as an information reservoir. This assumption is also compatible with biological examples. The main difficulty arises from the presence of the feedback, that is the signal influences the receptor and thus the readout population, which in turn impacts the storage population and finally changes the deactivation rate of the receptor - schematically, $H \to R \to U \to S \to R$, but the causal order does not reflect the temporal one.

We start with the master equation for the propagator $P(u, r, s, h, t|u_0, r_0, s_0, h_0, t_0)$,

$$\partial_t P = \left[ \frac{\hat{W}_U(r)}{\tau_U} + \frac{\hat{W}_R(s, h)}{\tau_R} + \frac{\hat{W}_S(u)}{\tau_S} + \frac{\hat{W}_H}{\tau_H} \right] P.$$

We rescale the time by $\tau_S$ and introduce two small parameters to control the timescale separation analysis, $\epsilon = \tau_U/\tau_R$ and $\delta = \tau_R/\tau_H$. Since $\tau_S/\tau_H = \mathcal{O}(1)$, we set it to 1 without loss of generality. We then write $P = P^{(0)} + \epsilon P^{(1)}$ and expand the master equation to find $P^{(0)} = p_{U|R}^{\text{st}}(u|r)\Pi$, with $\hat{W}_U p_{U|R}^{\text{st}} = 0$. We obtain that $\Pi$ obeys the following equation:

$$\partial_t \Pi = \left[ \delta^{-1} \hat{W}_R(s, h) + \hat{W}_S(u) + \hat{W}_H \right] \Pi.$$

Yet again, $\Pi = \Pi^{(0)} + \delta \Pi^{(1)}$ allows us to write $\Pi^{(0)} = p_{R|S,H}^{\text{st}}(r|s, h)F(s, h, t|s_0, h_0, t_0)$ at order $\mathcal{O}(\delta^{-1})$, where $\hat{W}_R p_{R|S,H}^{\text{st}} = 0$. Expanding first in $\epsilon$ and then in $\delta$ sets a hierarchy among timescales. Crucially, due to the feedback present in the system, we cannot solve the next order explicitly to find $F$. Indeed, after a marginalization over $r$, we find $\partial_t F = \left[ \hat{W}_H + \hat{W}_S \left( \bar{u}(s, h) \right) \right] F$, at order $\mathcal{O}(1)$, where $\bar{u}(s, h) = \sum_{u,r} u\, p_{U|R}^{\text{st}}(u|r) p_{R|S,H}^{\text{st}}(r|s, h)$. Hence, the evolution operator for $F$ depends manifestly on $s$, and the equation cannot be self-consistently solved. To tackle the problem, we first discretize time, considering a small interval, that is $t = t_0 + \Delta t$ with $\Delta t \ll \tau_U$ and thus $\bar{u}(s, h) \approx u_0$. We thus find $F(s, h, t|s_0, h_0, t_0) = P(s, t|s_0, t_0)P_H(h, t|h_0, t_0)$ in the domain $t \in [t_0, t_0 + \Delta t]$, since $H$ evolves independently from the system (see also Supplementary Information for analytical steps).

Iterating the procedure for multiple time steps, we end up with a recursive equation for the joint probability $p_{U,R,S,H}(u, r, s, h, t_0 + \Delta t)$. We are interested in the following marginalization

$$p_{U,S}(u, t+\Delta t) = \sum_{r=0}^{1} \int_0^\infty dh\, p_{u|R}^{st}(u|r)\, p_{R|S,H}^{st}(r|h, s)\, p_H(h, t+\Delta t) \sum_{s'=0}^{N_S} \sum_{u'=0}^{\infty} P(s', t \to s, t + \Delta t|u')\, p_{U,S}(u', s', t)$$

where $P(s', t \to s, t + \Delta t)$ is the propagator of the storage at fixed readout. This is the Chapman-Kolmogorov equation in the timescale separation approximation. Notice that this solution requires the knowledge of $p_{U,S}$ at the previous time step, and it has to be solved iteratively.

## Explicit solution for the storage propagator

To find a numerical solution to our system, we first need to compute the propagator $P(s_0, t_0 \to s, t)$. Formally, we have to solve the master equation

$$\partial_t P(s_0 \to s|u_0) = \Gamma_S^0 \Big[ e^{-\beta\sigma} u_0 P(s_0 \to s')\delta_{s',s-1}$$

$$+ s' P(s_0 \to s')\delta_{s',s+1}$$

$$- P(s_0 \to s')\delta_{s',s} \left( s' + e^{-\beta\sigma} u_0 \right) \Big]$$

where we used the shorthand notation $P(s_0 \to s) = (s_0, t_0 \to s, t)$. Since our formula has to be iterated for small timesteps, that is $t - t_0 = \Delta t \ll 1$, we can write the propagator as follows

$$P(s_0, t_0 \rightarrow s, t_0 + \Delta t | u_0) = p_{S|U}^{\mathrm{st}} + \sum_\nu w_\nu a^{(\nu)} e^{\lambda_\nu \Delta t}$$

where $w_\nu$ and $\lambda_\nu$ are respectively eigenvectors and eigenvalues of the transition matrix $\hat{W}_S(u_0)$,

$$\begin{aligned}
\left(\hat{W}_S(u_0)\right)_{ij} &= e^{-\beta\sigma} u_0 & \text{if } i = j+1 \\
\left(\hat{W}_S(u_0)\right)_{ij} &= j & \text{if } i = j - 1 \\
\left(\hat{W}_S(u_0)\right)_{ij} &= 0 & \text{otherwise}
\end{aligned}$$

and the coefficients $a^{(\nu)}$ are such that

$$p_{S|U}(s_0, t_0 \rightarrow s, t_0 + \Delta t | u_0) = p_{S|U}^{\mathrm{st}} + \sum_\nu w_\nu a^{(\nu)} = \delta_{s,s_0}.$$

Since eigenvalues and eigenvectors of $\hat{W}_S(u_0)$ might be computationally expensive to find, we employ another simplification. As $\Delta t \rightarrow 0$, we can restrict the matrix only to jumps to the $n$-th nearest neighbors of the initial state $(s_0, t_0)$, assuming that all other states are left unchanged in small time intervals. We take $n = 2$ and check the accuracy of this approximation against the full simulation for a limited number of timesteps.

## Mean-field relations

We note that $\langle U \rangle$ and $\langle S \rangle$ satisfies the following mean-field relationship:

$$\frac{\langle U \rangle - \langle U \rangle_{r=1}}{\langle U \rangle_{r=1} - \langle U \rangle_{r=0}} = f_0\left(\frac{\langle S \rangle}{N_S}\right), \tag{17}$$

where $f_0(x)$ is an analytical function of its argument (see Supplementary Information). This relation clearly states that only the fraction of active storage units is relevant to determining the habituation dynamics.

## Mutual information

Once we have $p_U(u, t)$ (obtained marginalizing $p_{U,S}$ over $s$) for a given $p_H(h, t)$, we can compute the mutual information

$$I_{U,H}(t) = \mathcal{H}[p_U](t) - \int_0^\infty dh\, p_H(h, t)\mathcal{H}[p_{U|H}](t)$$

where $\mathcal{H}$ is the Shannon entropy. For the sake of simplicity, we consider that the external signal follows an exponential distribution $p_H(h, t) = \lambda(t)e^{-\lambda(t)h}$. Notice that, in order to determine such quantity, we need the conditional probability $p_{U|H}(u, t)$. In the Supplementary Information, we show how all the necessary joint and conditional probability distributions can be computed from the dynamical evolution derived above.

We also highlight here that the timescale separation implies $I_{S,H} = 0$, since

$$p_{S|H}(s, t|h) = \sum_u p_{U,S|H}(u, s, t|h)$$

$$= p_S(s, t) \sum_u \sum_r p_{U|R}^{\mathrm{st}}(u|r)p_{R|S,H}^{\mathrm{st}}(r|s, h).$$

$$= p_S(s, t)$$

Although it may seem surprising, this is a direct consequence of the fact that $S$ is only influenced by $H$ through the stationary state of $U$. Crucially, the presence of the feedback is still fundamental in promoting habituation. Indeed, we can always write the mutual information between the signal $H$ and both the readout $U$ and the storage $S$ together as $I_{(U,S),H} = \Delta I_f + I_{U,H}$, where $\Delta I_f = I_{(U,S),H} - I_{U,H} = I_{(U,H),S} - I_{U,S}$. Since $\Delta I_f > 0$ (by standard information-theoretic inequalities), the storage is increasing the information of the two populations together on the external signal. Overall,

although $S$ and $H$ are independent in this limit, the feedback is paramount in shaping how the system responds to the external signal and stores information about it.

## Pareto optimization

We perform a Pareto optimization at stationarity in the presence of a prolonged stimulation. We seek the optimal values of $(\beta, \sigma)$ by maximizing the functional in *Equation 15* of the main text. Hence, we maximize the information between the readout and the signal, simultaneously minimizing the dissipation of the receptor induced by both the signal and feedback process and the dissipation associated with storage production, as discussed in the main text. The dissipative contributions have been computed per unit energy to be comparable with the mutual information. In the Supplementary Information, we detailed the derivation of the Pareto front and investigated the robustness of this optimization strategy.

## Recording of whole brain neuronal activity in zebrafish larvae

Acquisitions of the zebrafish brain activity were carried out in one Elavl3:H2BGCaMP6s larvae at 5 days post fertilization raised at 28 °C on a 12 hr light/12 hr dark cycle according to the approval by the Ethical Committee of the University of Padua (61/2020 dal Maschio). The subject was embedded in 2% agarose gel and brain activity was recorded using a multiphoton system with a custom 3D volumetric acquisition module. Data were acquired at 30 frames per second covering an effective field of view of about $450 \times 900\,\mathrm{um}$ with a resolution of 512×1024 pixels. The volumetric module acquires a volume of about $180 - 200\,\mathrm{um}$ in thickness encompassing 30 planes separated by about $7\,\mathrm{um}$, at a rate of 1 volume per second, sufficient to track the slow dynamics associated with the fluorescence-based activity reporter GCaMP6s. Visual stimulation was presented in the form of a looming stimulus with 150 s intervals, centered with the fish eye (see Supplementary Information). Neurons identification and anatomical registrations were performed as described in *Bruzzone et al., 2021*.

## Data analysis

The acquired temporal series were first processed using an automatic pipeline, including motion artifact correction, temporal filtering with a 3s rectangular window, and automatic segmentation. The obtained dataset was manually curated to resolve segmentation errors or to integrate cells not detected automatically. We fit the activity profiles of about 55,000 cells with a linear regression model using a set of base functions representing the expected responses to each stimulation event. These base functions have been obtained by convolving the exponentially decaying kernel of the GCaMP signal lifetime with square waveforms characterizing the presentation of the corresponding visual stimulus. The resulting score coefficients of the fit were used to extract the cells whose score fell within the top 5% of the distribution, resulting in a population of $\approx 2400$ neurons whose temporal activity profile correlates most with the stimulation protocol. The resulting fluorescence signals $F^{(i)}$ were processed by removing a moving baseline to account for baseline drifting and fast oscillatory noise (*Jia et al., 2011*). See Supplementary Information.

## Model for neural activity

Here, we describe how our framework is modified to mimic neural activity. Each readout unit, $u$, is interpreted as a population of $N$ neurons, i.e., a region dedicated to the sensing of a specific input. When a readout population is activated at time $t$, each of its $N$ neurons fires with a probability $p$. We set $N = 20$ and $p = 0.5$ has been set to have the same number of observed neurons in data and simulations, while $p$ only controls the dispersal of the points in *Figure 6c*, thus not altering the main message. The dynamics of each readout unit follows our dynamical model. Due to habituation, some of the readout units activated by the first stimulus will not be activated by subsequent stimuli. Although the evoked neural response cannot be captured by this extremely simple model, its archetypal ingredients (dissipation, storage, and feedback) are informative enough to reproduce the low-dimensional habituation dynamics found in experimental data.

## Acknowledgements

GN, SS, and DMB acknowledge Amos Maritan for fruitful discussions. DMB thanks Paolo De Los Rios for insightful comments. GN and DMB acknowledge the Max Planck Institute for the Physics of

Complex Systems for hosting GN during several stages of this work. SS acknowledges #NEXTGENER-ATIONEU (NGEU) and funding by the Ministry of Universities and Research (MUR), National Recovery and Resilience Plan (NRRP), project MNESYS (PE0000006) – A Multiscale integrated approach to the study of the nervous system in health and disease (DN 1553 11.10.2022). GN acknowledges funding provided by the Swiss National Science Foundation through its Grant CRSII5_186422. DMB is funded by the STARS@UNIPD grant with the project "ActiveInfo"

## Additional information

### Funding

| Funder | Grant reference number | Author |
| --- | --- | --- |
| Ministero dell'Università e della Ricerca | MNESYS PE0000006 | Samir Suweis |
| Swiss National Science Foundation | CRSII5_186422 | Giorgio Nicoletti |
| Max Planck Institute for the Physics of Complex Systems | | Daniel Maria Busiello |
| University of Padova | ActiveInfo | Daniel Maria Busiello |
| Ministero dell'Università e della Ricerca | #NEXTGENERATIONEU (NGEU) | Samir Suweis |

The funders had no role in study design, data collection and interpretation, or the decision to submit the work for publication. Open access funding provided by Max Planck Society.

### Author contributions

Giorgio Nicoletti, Conceptualization, Data curation, Formal analysis, Investigation, Visualization, Methodology, Writing – original draft, Writing – review and editing; Matteo Bruzzone, Marco dal Maschio, Data curation, Methodology, Writing – original draft, Writing – review and editing; Samir Suweis, Data curation, Visualization, Methodology, Writing – original draft, Writing – review and editing; Daniel Maria Busiello, Conceptualization, Formal analysis, Funding acquisition, Investigation, Visualization, Methodology, Writing – original draft, Writing – review and editing

### Author ORCIDs

Giorgio Nicoletti https://orcid.org/0000-0002-7682-0596
Matteo Bruzzone https://orcid.org/0000-0001-7683-8107
Samir Suweis https://orcid.org/0000-0002-1603-8375
Marco dal Maschio https://orcid.org/0000-0003-0150-6647
Daniel Maria Busiello https://orcid.org/0000-0002-6754-5019

### Ethics

Acquisitions of the zebrafish brain activity were carried out in one Elavl3:H2BGCaMP6s larvae at 5 days post fertilization raised at 28°C on a 12 h light/12 h dark cycle according to the approval by the Ethical Committee of the University of Padua (61/2020 dal Maschio).

Reviewer #2 (Public review): https://doi.org/10.7554/eLife.99767.3.sa1
Reviewer #3 (Public review): https://doi.org/10.7554/eLife.99767.3.sa2
Author response https://doi.org/10.7554/eLife.99767.3.sa3

## Additional files

### Supplementary files

MDAR checklist

## Data availability

The data to produce *Figure 6* have been deposited on Zenodo and are accessible through the following link: https://doi.org/10.5281/zenodo.15683642.

The following dataset was generated:

| Author(s) | Year | Dataset title | Dataset URL | Database and Identifier |
|---|---|---|---|---|
| Matteo B, Marco DM, Giorgio N | 2025 | Habituation during visual stimulation in zebrafish brain activity | https://doi.org/10.5281/zenodo.15683642 | Zenodo, 10.5281/zenodo.15683642 |

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

## Appendix 1

### Detailed solution of the master equation

Consider the transition rates introduced in the main text:

$$\Gamma_{P \to A}^{(H)} = e^{\beta(h - \Delta E)}\Gamma_H^0 \qquad \Gamma_{A \to P}^{(H)} = \Gamma_H^0$$
$$\Gamma_{P \to A}^{(I)} = e^{-\beta \Delta E}\Gamma_I^0 \qquad \Gamma_{A \to P}^{(I)} = \Gamma_I^0 e^{\beta \kappa \sigma s / N_s}$$
$$\Gamma_{u \to u+1} = e^{-\beta(V - cr)}\Gamma_U^0 \qquad \Gamma_{u+1 \to u} = (u+1)\Gamma_U^0$$
$$\Gamma_{s \to s+1} = e^{-\beta \sigma} u \Gamma_S^0 \qquad \Gamma_{s+1 \to s} = (s+1)\Gamma_S^0.$$

We set a reflective boundary for the storage at $s = N_S$, corresponding to the maximum amount of storage molecules in the system. Moreover, for the sake of simplicity, we take $\Gamma_I^0 = \Gamma_H^0 \equiv \Gamma_R^0$. Retracing the steps of the Materials and methods, the master equation governing the evolution of the propagator of all variables, $P(u, r, s, h, t | u_0, r_0, s_0, h_0, t_0)$, is:

$$\partial_t P = \left[ \frac{\hat{W}_U(r)}{\tau_U} + \frac{\hat{W}_R(s, h)}{\tau_R} + \frac{\hat{W}_S(u)}{\tau_S} + \frac{\hat{W}_H}{\tau_H} \right] P. \tag{A1 - 1}$$

We solve this equation employing a timescale separation, i.e., $\tau_U \ll \tau_R \ll \tau_S \sim \tau_H$, where $\tau_X = \Gamma_X^0$ for $X = U, R, S$ and $\tau_H$ is the typical timescale of the signal dynamics. Motivated by several biological examples, we assumed that the readout population undergoes the fastest dynamics, while storage and signal evolution are the slowest ones. Defining $\epsilon = \tau_U / \tau_R$ and $\delta = \tau_R / \tau_H$, and setting $\tau_S / \tau_H = 1$ without loss of generality, we have:

$$\partial_t P = \left[ \epsilon^{-1} \delta^{-1} \hat{W}_U(r) + \delta^{-1} \hat{W}_R(s, h) + \hat{W}_S(u) + \frac{\tau_S}{\tau_H} \hat{W}_H \right] P. \tag{A1 - 2}$$

We propose a solution in the following form, $P = P^{(0)} + \epsilon P^{(1)}$. By inserting this expression in the equation above, and solving order by order in $\epsilon$, at order $\epsilon^{-1}$, we have that:

$$P^{(0)} = P_{U|R}^{\mathrm{st}}(u \mid r)\Pi(r, h, t \mid r_0, h_0, t_0) \tag{A1-3}$$

where $p^{\mathrm{st}}$ solves the master equation for the readout evolution at a fixed $r$:

$$0 = p_{U|R}^{\mathrm{st}}(u + 1)[u + 1] + p_{U|R}^{\mathrm{st}}\alpha(r) - p_{U|R}^{\mathrm{st}}(u)[u + \alpha(r)] \tag{A1 - 4}$$

with $\alpha(r) = e^{-\beta(V - cr)}$. Hence,

$$p_{U|R}^{\mathrm{st}}(u|r) = e^{-\alpha(r)} \frac{\alpha(r)^u}{u!}. \tag{A1 - 5}$$

At order $\epsilon^0$, we find the equation for $\Pi$, also reported in the Materials and methods:

$$\partial_t \Pi(r, h, t | r_0, h_0, t_0) = \left[ \delta^{-1} \hat{W}_R(h) + \hat{W}_S(u) + \hat{W}_H \right] \Pi(r, h, t | r_0, h_0, t_0). \tag{A1 - 6}$$

To solve this equation, we propose a solution of the form $\Pi = \Pi^{(0)} + \delta \Pi^{(1)}$. Hence, again, at order $\delta^{-1}$, we have that $\Pi^{(0)} = p_{R|S,H}^{\mathrm{st}}(r|s, h)F(s, h, t|s_0, h_0, t_0)$, where $p_{R|S,H}^{\mathrm{st}}$ satisfy the steady-state equation for the fastest degree of freedom, with all the others fixed. In the case, it is just the solution of the rate equation for the receptor:

$$p_{R|H,S}^{\mathrm{st}}(r = 1) = \frac{\Gamma_{P \to A}^{\mathrm{eff}}}{\Gamma_{P \to A}^{\mathrm{eff}} + \Gamma_{A \to P}^{\mathrm{eff}}}, \quad p_{R|H}^{\mathrm{st}}(r = 0) = 1 - p_{R|H}^{\mathrm{st}}(r = 1, t) \tag{A1 - 7}$$

where $\Gamma_{P \to A}^{\mathrm{eff}} = \Gamma_{P \to A}^{(I)} + \Gamma_{P \to A}^{(H)}$, and the same for the reverse reaction. At order $\delta^{-1}$, we have an equation for $F$:

$$\partial_t F(s, h, t|s_0, h_0, t_0) = \sum_{r,u} p_{U|R}^{\text{st}}(u|r) \left[ \hat{W}_S(u) + \hat{W}_H \right] \left[ p_{R|S,H}^{\text{st}}(r|s, h)F(s, h, t|s_0, h_0, t_0) \right] \tag{A1 - 8}$$

As already explained in the Materials and methods, due to the feedback, this equation cannot be solved explicitly. Indeed, the operator governing the evolution of F is:

$$\hat{W}_{\text{eff}} = \hat{W}_H + \sum_u p_{U|S,H}^{\text{st}}(u|s, h)\hat{W}_S(u) = \hat{W}_H + \hat{W}_S \left( \sum_u u\, p_{U|S,H}^{\text{st}}(u|s, h) \right) = \hat{W}_H + \hat{W}_S \left( \bar{u}(s, h) \right) \tag{A1 - 9}$$

with $p_{U|S,H}^{\text{st}}(u|s, h) = \sum_r p_{U|R}^{\text{st}}(u|r)p_{R|S,H}^{\text{st}}(r|s, h)$ and using the linearity of $\hat{W}_S(u)$. In order to solve this equation, we shall assume that $\bar{u}(s, h) = u_0$, bearing in mind that this approximation holds if $t$ is small enough, that is $t = t_0 + \Delta t$ with $\Delta t \ll \tau_U$. Therefore, for a small interval, we have:

$$\partial_t F(s, h, t_0 + \Delta t|s_0, h_0, t_0) = \left[ \hat{W}_S(u_0) + \hat{W}_H \right] F(s, h, t|s_0, h_0, t_0) \tag{A1 - 10}$$

Overall, we end up with the following joint probability of the model at time $t_0 + \Delta t$

$$p_{U,R,S,H}(u, r, s, h, t_0 + \Delta t)$$
$$= \sum_{u_0,s_0} P_{U|R}^{\text{st}}{}^{(u|r)} P_{R|s,h}^{\text{st}}(r \mid s, h)P(s, t_0 + \Delta t \mid s_0, u_0, t_0) \int dh_0 P_H(h, t_0 + \Delta t \mid h, T_0)P_{U,S,H}(u_0, s_0, h_0, t_0)$$
$$= \sum_{u_0,s_0} P_{U|R}^{\text{st}}(u|r)P_{R|s,h}^{\text{st}}(r \mid s, h)P(s, t_0 + \Delta t \mid s_0, u_0, t_0)p_{U,S}(u_0, s_0, t_0)p_H(h, t_0 + \Delta t) \tag{A1-11}$$

where $\int dh_0 P_H(h, t_0 + \Delta t|h_0, t_0)p_{U,S,H}(u_0, s_0, h_0, t_0) = p_{U,S}(u_0, s_0, t_0)p_H(h, t_0 + \Delta t)$ since $H$ at time $t_0 + \Delta t$ is independent of $S$ and $U$. When propagating the evolution through intervals of duration $\Delta t$, we also assume that $H$ evolves independently since it is an external variable, while affecting the evolution of the other degrees of freedom. This structure reflects into the equation above. For simplicity, we prescribe $p_H(h, t)$ to be an exponential distribution, $p_H(h, t) = \lambda(t)e^{-\lambda(t)h}$, and solve iteratively **Equation A1-11** from $t_0$ to a given $T$ in steps of duration $\Delta t$, as indicated above. This complex iterative solution arises from the timescale separation because of the cyclic feedback structure: $\{S, H\} \to R \to U \to S$. This solution corresponds explicitly to

$$p_{U,S}(u, t + \Delta t) = \sum_{r=0}^{1} \int_0^\infty dh\, p_{U|R}^{\text{st}}(u|r)\, p_{R|S,H}^{\text{st}}(r|h, s)\, p_H(h, t + \Delta t)$$
$$\sum_{s'=0}^{N_S} \sum_{u'=0}^{\infty} P(s', t \to s, t + \Delta t|u')p_{U,S}(u', s', t) \tag{A1-12}$$

where $P(s', t \to s, t + \Delta t)$ is the propagator of the storage at fixed readout. This is the Chapman-Kolmogorov equation in the time-scale separation approximation. Notice that this solution requires the knowledge of $p_{U,S}$ at the previous time step, and it has to be solved iteratively. Both $p_U$ and $p_S$ can be obtained by an immediate marginalization.

As detailed in the Materials and methods, the propagator $P(s_0, t_0 \to s, t)$, when restricted to small time intervals, can be obtained by solving the birth-and-death process for storage molecules at fixed readout, limiting the state space only to $n$ nearest neighbors (we checked that our results are robust increasing $n$ for the selected simulation time step).

## Information-theoretic quantities

By direct marginalization of **Equation A1-12**, we obtain the evolution of $p_U(u, t)$ and $p_S(s, t)$ for a given $p_H(h, t)$. Hence, we can compute the mutual information as follows:

$$I_{U,H}(t) = \mathcal{H}[p_U](t) - \int_0^\infty dh\, p_H(h, t)\mathcal{H}[p_{U|H}](t) = -\frac{\Delta \mathbb{S}_U}{k_B} \tag{A1-13}$$

where $\mathcal{H}[p_X]$ is the Shannon entropy of $X$, and $\Delta \mathbb{S}_U$ is the reduction in the entropy of $U$ due to repeated measurement (see main text). Notice that, in order to determine such quantity, we need the conditional probability $p_{U|H}(u, t)$. This distribution represents the probability that, at a given time, the system jumps at a value $u$ in the presence of a given signal $h$. In order to compute it, we can write

$$p_{U|H}(u, t + \Delta t) = \sum_{s=0}^{M_S} \sum_{r=0}^{1} p_{U|R}^{\text{st}}(u|r) \, p_{R|S,H}^{\text{st}}(r|h, s) \, p_S(s, t + \Delta t) \tag{A1 - 14}$$

by definition. The only dependence on $h$ enters in $p_{R|S,H}^{\text{st}}$ through the $e^{\beta h}$ dependence in the rates.

Analogously, all the other mutual information can be obtained. As we showed in the Materials and methods, although $I_{S,H} = 0$ due to the time-scale separation, the presence of the feedback is still fundamental to effectively process information about the signal. This effect can be quantified through the feedback information $\Delta I_f = I_{(U,S),H} - I_{U,H} > 0$, as it captures how much the knowledge of $S$ and $U$ together helps to encode information about the signal with respect to $U$ alone. In terms of system entropy, we equivalently have:

$$k_B \Delta I_f = -\Delta \mathbb{S}_{U,S} + \Delta \mathbb{S}_U > 0 \tag{A1 - 15}$$

that highlights how much the effect of $S$ (feedback) reduces the entropy of the system due to repeated measurements. In practice, in order to evaluate $I_{(U,S),H}$, we exploit the following equality:

$$I_{(U,S),H} = \mathcal{H}\left[p_{U,S}\right](t) - \int_0^\infty dh p_H(h, t) \mathcal{H}\left[p_{U,S|H}\right](t). \tag{A1 - 16}$$

for which we need $p_{U,S|H}$. It can be obtained by noting that

$$p_{U,S}(u, s, t) = p_{U|S}(u, t|s) p_S(s, t) = \int dh \sum_r p_{U|R}^{\text{st}}(u|r) p_{R|S,H}^{\text{st}}(r|s, h) p_S(s, t) p_H(h, t) \tag{A1-17}$$

from which we immediately see that

$$p_{U,S|H}(u, s, t) = \sum_{r=0}^{1} p_{U|R}^{\text{st}}(u|r) \, p_{R|S,H}^{\text{st}}(r|h, s) p_S(s, t) \tag{A1 - 18}$$

that can be easily computed at any given time $t$.

## Mean-field relation between average readout and storage

Fixing all model parameters, the average value of storage, $\langle S \rangle$, and readout, $\langle U \rangle$, is numerically determined by solving iteratively the system, as shown above. However, an analytical relation between these two quantities can be found starting from the definition of $\langle U \rangle$:

$$\langle U \rangle = \sum_{u,s} u P_{U,S}^{\text{st}}(u, s) = \sum_{u,s} u P_{U|S}^{\text{st}}(u|s) P_S^{\text{st}}(s) = \sum_{u,s,r} u P_{U|R}^{\text{st}}(u|r) P_{R|S}^{\text{st}}(r|s) P_S^{\text{st}}(s) \tag{A1 - 19}$$

Then, inserting the expression for the stationary probability that we know analytically:

$$\langle U \rangle = \sum_s \left( \langle u P_{U|R}^{\text{st}}(u|r = 0) \rangle P_{R|S}^{\text{st}}(r = 0|s) + \langle u P_{U|R}^{\text{st}}(u|r = 1) \rangle P_{R|S}^{\text{st}}(r = 1|s) \right) P_S^{\text{st}}(s) \tag{A1 - 20}$$

where $P_{R|S}^{\text{st}} = \int dh P_{R|H,S}^{\text{st}} P_H dh \equiv f_R(\rho_S)$ has a complicated expression involving the hypergeometric function $_2F_1$ in terms of model parameters and only the fraction of active $S$, $\rho_S = s/N_S$ (the explicit derivation of this formula is not shown here). Then, we have:

$$\langle U \rangle = \sum_s \left( e^{-\beta V} f_0(\rho_S) + e^{-\beta(V-c)}(1 - f_0(\rho_S)) \right) P_S^{\text{st}}(s) \tag{A1 - 21}$$

Since we do not have an analytical expression for $P_S^{\text{st}}(s)$, we employ the mean-field approximation, reducing all the correlation functions to products of averages:

$$\langle U \rangle = e^{-\beta(V-c)} + e^{-\beta V} f_0(\bar{\rho}_S) \left( 1 - e^{\beta c} \right) \tag{A1 - 22}$$

where $\bar{\rho}_S = \langle S \rangle / N_S$. This clearly shows that, given a set of model parameters, $\langle U \rangle$ and the average fraction of storage molecules, $\bar{\rho}_S$ are related. In particular, introducing the change of parameters presented in the Materials and methods, we have the following collapse:

$$\frac{\langle U \rangle - \langle U \rangle_A}{\langle U \rangle_A - \langle U \rangle_P} = f_0(\bar{\rho}_S) \tag{A1 - 23}$$

where $\langle U \rangle_A$ and $\langle U \rangle_P$ are respectively the average of $U$ fixing $r = 1$ (active receptor) and $r = 0$ (passive receptor). It is also possible to perform an expansion of $f_0$ which numerically results in being very precise:

$$\frac{\langle U \rangle - \langle U \rangle_A}{\langle U \rangle_A - \langle U \rangle_P} = \frac{a_{-1}(\lambda_H, \beta, g)}{z} + a_0(\lambda_H, \beta) + a_1(\lambda_H, \beta, g)z^{-1-\lambda_H/\beta} + a_2(\lambda_H, \beta)z^{-\lambda_H/\beta} \tag{A1 - 24}$$

where $z = e^{\beta \Delta E}\left(1 + g\, e^{\beta \bar{\rho}_S/\alpha \lambda_H}\right)$. Since all these relations just depend on the average fraction of storage molecules, it is natural to ask what happens when $N_S \to N_S' = nN_S$. Fixing all the remaining parameters, both $\langle U \rangle$ and $\bar{\rho}_S$ will change, still satisfying the mutual relation presented above. Let us consider, for $N_S'$, the stationary solution that has the same fraction of $S$, i.e., $(\bar{\rho}_S)_{N_S'} = (\bar{\rho}_S)_{N_S}$. As a consequence of the scaling relation, $\langle U \rangle_{N_S'} \neq \langle U \rangle_{N_S}$. Considering $\langle U \rangle_P \approx 0$ in both settings, we can ask ourselves what is the factor $\gamma$ such that $\gamma(\langle U \rangle_A)_{N_S} = (\langle U \rangle_A)_{N_S'}$. Since $u$ only enters linearly in the dynamics of the storage, and the mutual relation only depends on the fraction of active $S$, we guess that $\gamma = 1/n$, as numerically observed. As stated in the main text, we can finally conclude that the storage fraction is the most relevant quantity in our model to determine the effect of the feedback and characterize the dynamical evolution. This observation makes our conclusions more robust, as they do not depend on the specific choice for the storage reservoir since there always exists a scaling relation connecting $\langle U \rangle$ and $\bar{\rho}_S$. As such, changing the value of the model parameters we fixed will only affect the number of active molecules without modifying the main results presented in this work.

**Appendix 1—table 1.** Summary of the model parameters and the values used for numerical simulations, unless otherwise specified.

The parameters $\beta$ and $\sigma$ qualitatively determine the behavior of the model and are varied.

| Model parameter | Description | Typical value |
| --- | --- | --- |
| $M_S$ | Maximum number of storage units | 30 |
| $\Delta E$ | Receptor energetic barrier | 1 |
| $\langle U \rangle_P$ | Average readout with passive receptor | 150 |
| $\langle U \rangle_A$ | Average readout with active receptor | 0.5 |
| $\Gamma_S^0$ | Inverse timescale of the storage | 1 |
| $g$ | Receptor's pathways timescale ratio | 1 |
| $\alpha$ | Inhibiting storage fraction | 2/3 |
| $H_{\text{ref}}$ | Reference signal | 10 |
| $\langle H \rangle_{\text{max}}$ | Average signal strength | 10 |
| $\langle H \rangle_{\text{min}}$ | Average background strength | 0.1 |
| $\Delta T$ | Duration of the pause between two signals | 100 |
| $T_{\text{on}}$ | Duration of a signal | 100 |
| $\Delta t$ | Timestep used in simulations | 5.1 |
| $\beta$ | Inverse temperature | - |
| $\sigma$ | Storage energy cost | - |

## Appendix 2

## Model features and robustness of optimality

In this section, we show how different choices of model parameters and the external signal features impact the results presented in the main text. In *Appendix 1—table 1* we summarize for convenience the parameters of the model. We recall that, for analytical ease, we take the environment to be an exponentially distributed signal,

$$p_H(h, t) = \lambda(t)e^{-h\lambda(t)} \tag{A2 - 1}$$

where $\lambda$ is its inverse characteristic scale. In particular, we describe the case in which no signal is present by setting $\lambda$ to be large, so that the typical realizations of $H$ would be too small to activate the receptors. On the other hand, when $\lambda$ is small, the values of $h$ appearing in the rates of the model are large enough to activate the receptor and thus allow the system to sense the signal. In the dynamical case, we take $\lambda(t)$ to be a square wave, so that $\langle H \rangle = 1/\lambda$ alternates between two values $\langle H \rangle_{\min}$ - the input signal - and $\langle H \rangle_{\max}$ - the background. We denote with $T_{\text{on}}$ the duration of $\langle H \rangle_{\max}$, and with $\Delta T$ the one of $\langle H \rangle_{\min}$, that is the pause between two subsequent signals. In practice, this mimics an on-off dynamics, where the stochastic signal is present when its average is $\langle H \rangle_{\max}$.

### Effects of the external signal strength and thermal noise level

In *Appendix 2—figure 1a*, we study the behavior of the model in the presence of a static exponential signal, with average $\langle H \rangle$. We focus on the case of low $\sigma$, so that the production of storage is favored. As $\langle H \rangle$ decreases, $I_{U,H}$ decreases as well. Hence, as expected, information acquired through sensing depends on the strength of the external signal that coincides with the energy input driving receptor activation. However, the system does not display for all parameters an emergent information dynamics, memory, and habituation. In *Appendix 2—figure 1b*, we see that, when the temperature is low but $\sigma$ is high, the system does not show habituation and $\Delta I_{U,H} = 0$. On the other hand, when thermal noise dominates (*Appendix 2—figure 1c*), even when the external signal is small, the system produces a large readout population due to random thermal activation. As a consequence, these random activations hinder the signal-driven ones, thus the system does not effectively sense the external signal even when present and $I_{U,H}$ is always small. It is important to remind here that, as we see in the main text, $I_{U,H}$ is not monotonic at fixed $\sigma$ and as a function of $\beta$. This is due to the fact that low temperatures typically favor sensing and habituation, but they also intrinsically suppress readout production. Thus, at high $\beta$, $\sigma$ needs to be small to effectively store information since thermal noise is negligible. Vice versa, a small $\sigma$ is detrimental at high temperatures since the system produces storage as a consequence of thermal noise. This complex interplay is captured by the Pareto optimization, which gives us an effective relation between $\beta$ and $\sigma$ to maximize storage while minimizing dissipation.

### Stationary behavior of the model with a constant signal

In this section, we detail the behavior of the model when exposed to a static signal. As in the main text, we take

$$p_H(h) = \lambda^{\text{st}}e^{-h\lambda^{\text{st}}} \tag{A2 - 2}$$

with $\langle H \rangle = 1/\lambda^{\text{st}} = H^{\text{st}}$.

We first consider the case where the system does not adapt its inhibition strength $\kappa$, that is we set

$$\kappa = \frac{H_{\text{ref}}}{\alpha\,\sigma} \tag{A2 - 3}$$

where $H_{\text{ref}}$ is the reference signal, and $\alpha$ the fraction of storage population needed to inhibit the receptor on average (see *Table 1*). In *Appendix 2—figure 2*, we plot as a function of $\beta$ and $\sigma$ the behavior of the stationary average readout population $\langle U \rangle^{\text{st}}$, the average storage population $\langle S \rangle^{\text{st}}$, the mutual information between readout and the signal $I_{U,H}^{\text{st}}$, and the total energy consumption $\delta Q_R^{\text{st}} + E_{\text{int}}^{\text{st}}$, where

$$E_{\text{int}}^{\text{st}} = \tau_S J_{\text{int}}^{\text{st}}/\sigma = \tau_S \sum_{u,s} \left[ \Gamma_{s \to s+1}\, p_{U,S}^{\text{st}}(u, s) - \Gamma_{s+1 \to s}\, p_{U,S}^{\text{st}}(u, s + 1) \right]$$

$$\tag{A2 - 4}$$

with $\tau_S = 1/\Gamma_S^0$. As shown in the main text, however, we can achieve a collapse of the Pareto fronts at different external signals if we allow the system to tune the inhibition strength as

$$\kappa(\langle H \rangle) = \frac{\langle H \rangle}{\alpha \, \sigma} \tag{A2-5}$$

so that a stronger input will correspond to a larger $\kappa$, and thus a stronger inhibition. In **Appendix 2—figure 3**, we show the behavior of the same stationary quantities in this case, and for a large range of $H^{\text{st}}$.

## Static and dynamical optimality

We now study the dynamical behavior of the model under a repeated external signal, for different values of $\langle H \rangle_{\text{max}}$. In particular, given an observable $O$, we define its change under a repeated signal, $\Delta O$, as the difference between the maximal response to the signal after several repetitions, once the system has habituated, and the maximal response to the first signal. In **Appendix 2—figure 4** we plot, as a function of $\beta$ and $\sigma$, the mutual information gain $\Delta I_{U,H}$, the feedback information gain $\Delta\Delta I_f$, the habituation strength $\Delta \langle U \rangle$, and the change in the internal energy flux $\Delta J_{\text{int}}$, when as before $\kappa$ is fixed by a reference signal $H_{\text{ref}}$. As in the main text, we see in particular that $\Delta \langle U \rangle$ is maximal in the region where the change in the mutual information $\Delta I_{U,H}$ and the feedback information $\Delta\Delta I_f$ are both small, suggesting that a strong habituation fueled by a large number of storage molecules with low energy cost is ultimately detrimental for information processing. Furthermore, in this region, the change in the internal energy flux, $J_{\text{int}}$, is large. For completeness, in **Appendix 2—figure 5** we plot all relevant dynamical quantities at different signal strength $\langle H \rangle_{\text{max}}$ in the case of a fixed $\kappa$ with a reference signal (**Equation 3**), whereas in **Appendix 2—figure 6** we focus on an adaptive $\kappa$ (**Equation A2-5**).

### Interplay between information storage and signal duration

In the main text and insofar, we have always considered the case $T_{\text{on}} = \Delta t$. We now study the effect of the signal duration and the pause length on sensing (**Appendix 2—figure 7**). If the system only receives short signals between long pauses, the slow storage build-up does not reach a high level of fraction of active molecules. As a consequence, the negative feedback on the receptor is less effective and habituation is suppressed (**Appendix 2—figure 7a**). Therefore, the peak of $\Delta I_{U,H}$ in the $(\beta, \sigma)$ plane takes place below the optimal curve, as $\sigma$ needs to be smaller than in the static case to boost storage production during the brief periods in which the signal is present. On the other hand, in **Appendix 2—figure 7b** we consider the case of a long signal with short pauses. In this scenario, the slow dynamical evolution of the storage can reach large values of number of molecules at larger values of $\sigma$, thus moving the optimal dynamical region slightly above the Pareto-like curve. The case of a short signal is comparable to the durations of the looming stimulations in the experimental setting, which can be used to tune the parameters of the model to the peak of information gain.

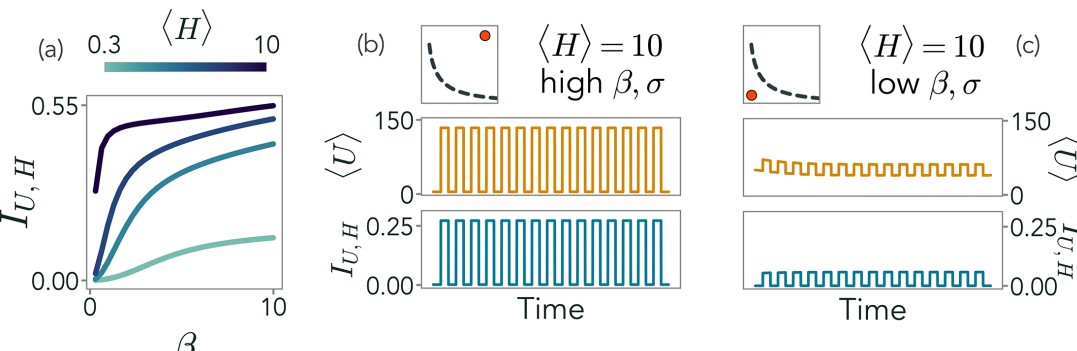

**Appendix 2—figure 1.** Effects of the external signal strength and thermal noise level on sensing. (**a**) At fixed $\sigma$ = 0.1 and constant $\langle H \rangle$, the system captures less information as $\langle H \rangle$ decreases and it needs to operate at high $\beta$ to sense the signal. In particular, as $\beta$ increases, $I_{U,H}$ becomes larger. (**b**) In the dynamical case, outside the optimal curve (black dashed line), at high $\beta$ and high $\sigma$, storage is not produced and no negative feedback is present. The
*Appendix 2—figure 1 continued on next page*

*Appendix 2—figure 1 continued*

system does not display habituation, and $I_{U,H}$ is smaller than on the optimal curve. (**c**) In the opposite regime, at low $\beta$ and $\sigma$, the system is dominated by thermal noise. As a consequence, the average readout $\langle U \rangle$ is high even when the external signal is not present ($\langle H \rangle = \langle H \rangle_{\min} = 0.1$), and it captures only a small amount of information $I_{U,H}$, which is masked by thermal activation. Simulation parameters are as in *Appendix 1—table 1*.

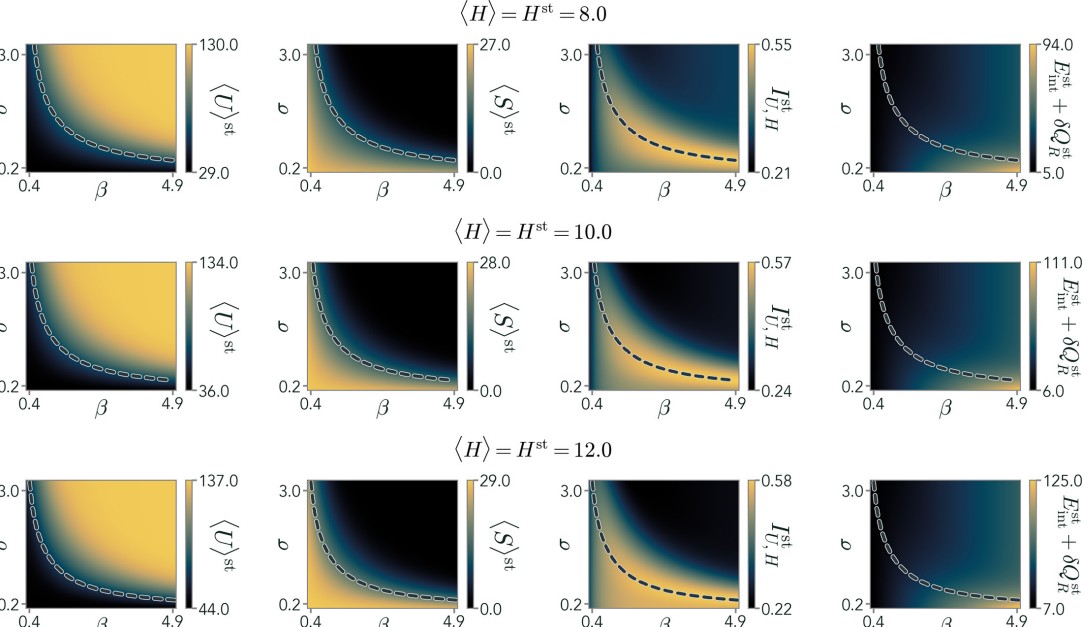

**Appendix 2—figure 2.** Behavior of the stationary average readout population $\langle U \rangle^{st}$, the average storage population $\langle S \rangle^{st}$, the mutual information between readout and the signal $I^{st}_{U,H}$, and the total energy consumption $\delta Q^{st}_R + E^{st}_{int}$, as a function of $\beta$ and $\sigma$ and in the presence of a static signal with average $H^{st}$. The value of $\kappa$ is fixed by a reference signal as in *Equation 2*. The dashed black line indicates the corresponding Pareto front. Simulation parameters are as in *Appendix 1—table 1*.

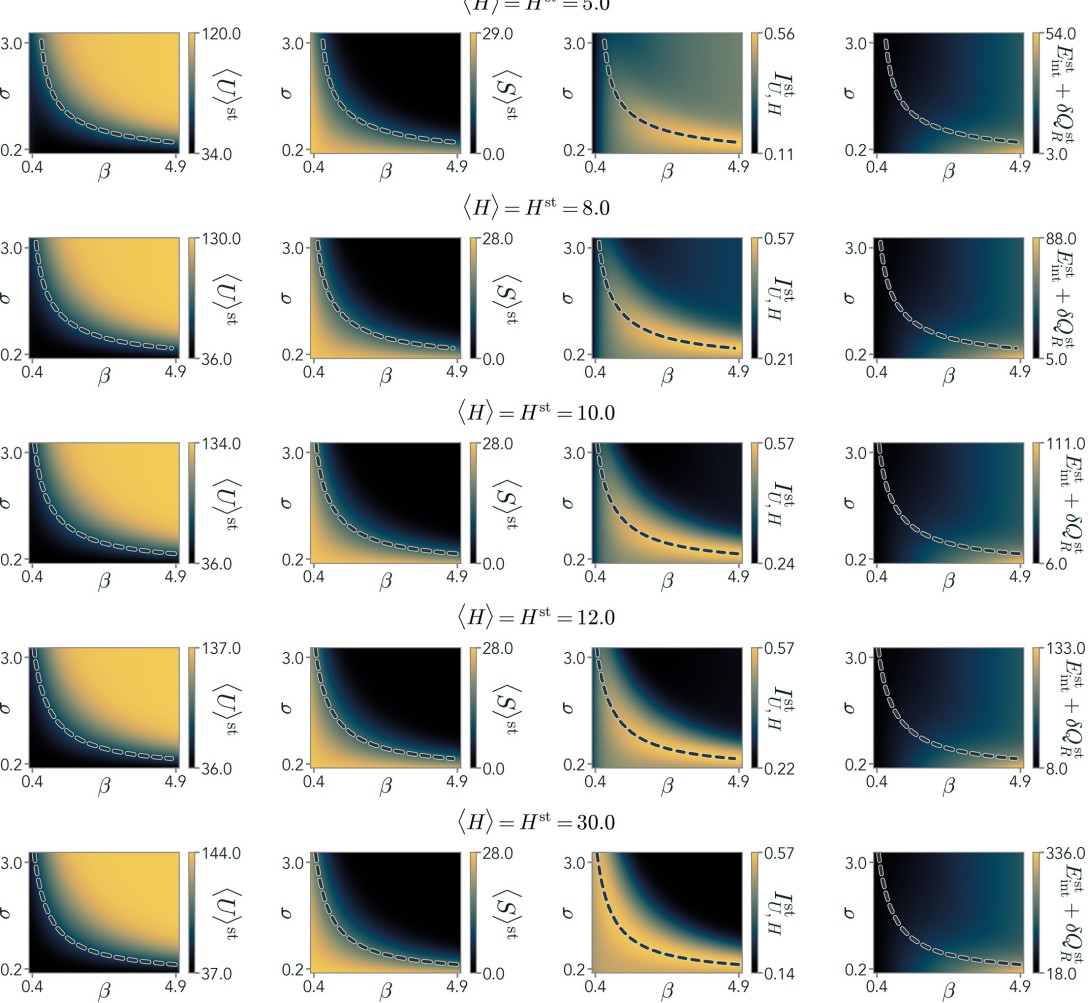

**Appendix 2—figure 3.** Behavior of the stationary average readout population $\langle U \rangle$st , the average storage population $\langle S \rangle$st , the mutual information between readout and the signal $I$st$U,H$, and the total energy consumption $\delta Q$st$R+$ $E$stint , as a function of $\beta$ and $\sigma$ and in the presence of a static signal with average $H$st . The value of $\kappa$ is tuned so that it follows the average value of the external signal, *Equation A2-5*. The dashed black line indicates the corresponding Pareto front. Simulation parameters are as in *Appendix 1—table 1*.

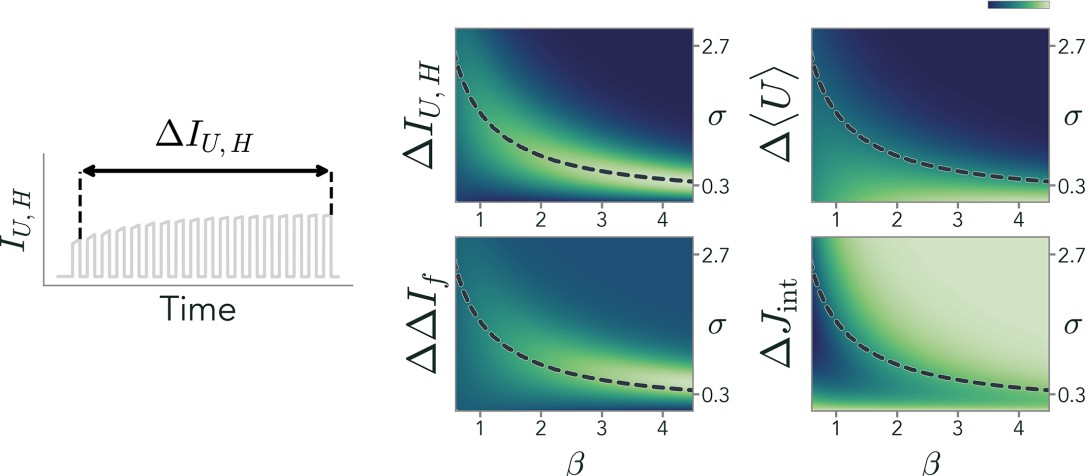

**Appendix 2—figure 4.** Dynamical optimality under a repeated external signal. (**a**) Schematic definition of how we study the dynamical evolution of relevant observables, by comparing the maximal response to a first signal with the one to a signal after the system has habituated. (**b**) Behavior of the increase in readout information, $\Delta I_{U,H}$, in feedback information, $\Delta\Delta I_f$, in average readout population, $\Delta\langle U\rangle$, and in the internal energy flux, $\Delta J_{int}$. The value of $\kappa$ is fixed by a reference signal as in *Equation 2*. The dashed black line indicates the corresponding Pareto front. Simulation parameters are as in *Appendix 1—table 1*.

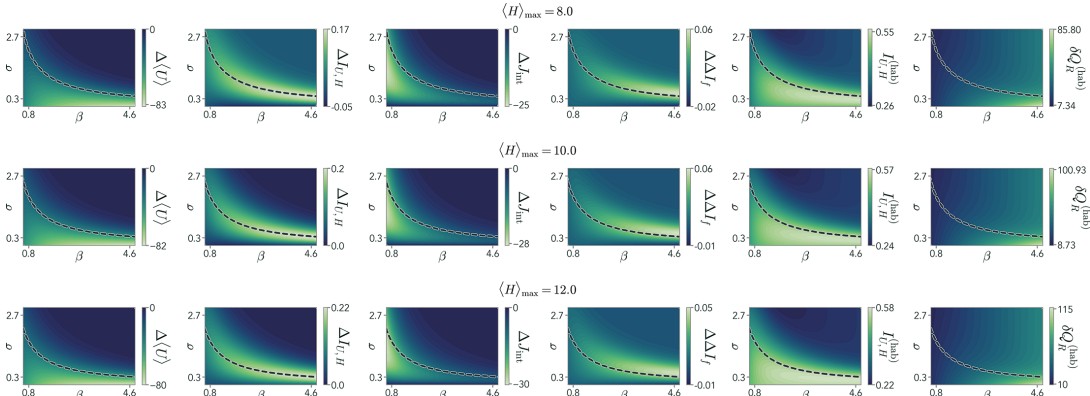

**Appendix 2—figure 5.** Behavior of the change in average readout population $\Delta\langle U\rangle$, readout information gain $\Delta I_{U,H}$, change in internal energy flux $\Delta J_{int}$, feedback information gain, $\Delta\Delta I_f$, final readout information after habituation $I^{(hab)}_{U,H}$, as a function of $\beta$ and $\sigma$ and in the presence of a switching signal with average $\langle H\rangle_{max}$. The value of $\kappa$ is fixed by a reference signal as in *Equation 2*. The dashed black line indicates the corresponding Pareto front. Simulation parameters are as in *Appendix 1—table 1*.

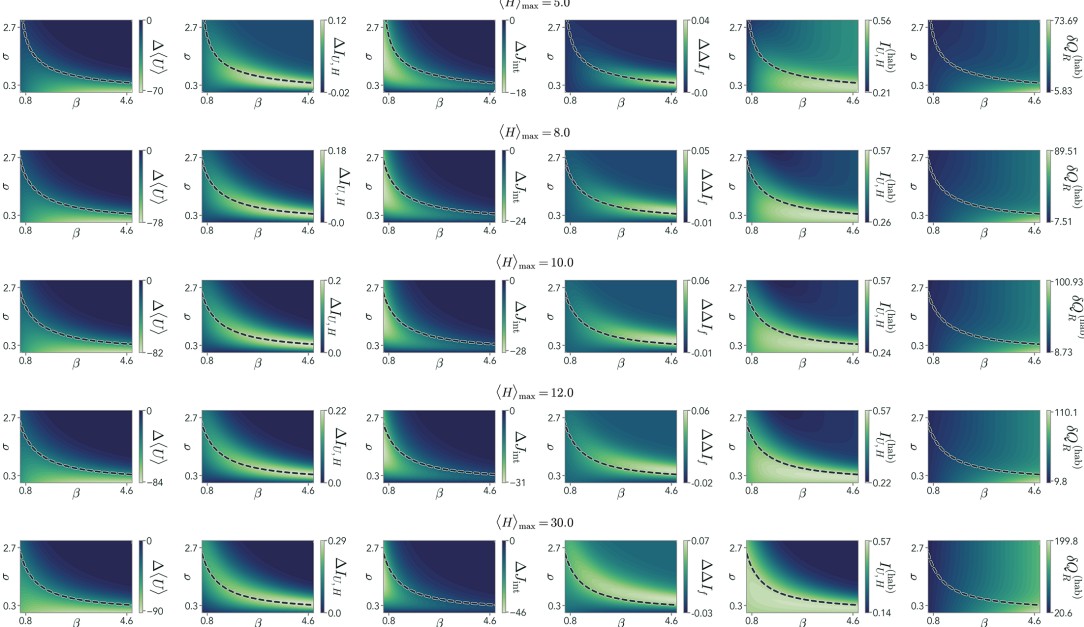

**Appendix 2—figure 6.** Behavior of the change in average readout population $\Delta\langle U\rangle$, readout information gain $\Delta I_{U,H}$, change in internal energy flux $\Delta J_{int}$, feedback information gain, $\Delta\Delta I_f$, final readout information after habituation $I^{(hab)}_{U,H}$, as a function of $\beta$ and $\sigma$ and in the presence of a switching signal with average $\langle H\rangle_{max}$. The value of $\kappa$ is tuned so that it follows the average value of the external signal as in *Equation A2-5*. The dashed black line indicates the corresponding Pareto front. Simulation parameters are as in *Appendix 1—table 1*.

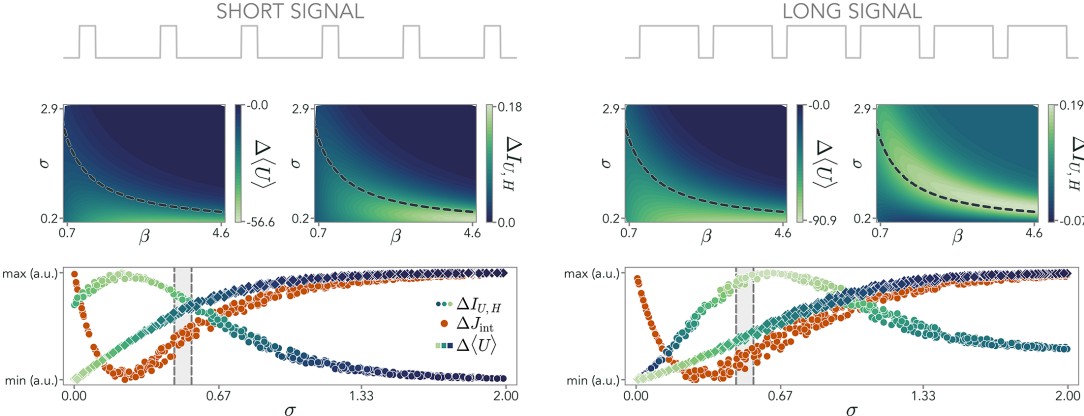

**Appendix 2—figure 7.** Effect of the signal duration on habituation. (**a**) If the system only receives the signal for a short time ($T_{on} = 50\Delta t < \Delta T = 200\Delta t$), it does not have enough time to reach a high level of storage molecules. As a consequence, both $\Delta U$ and $\Delta I_{U,H}$ are smaller, and thus habituation is less effective. (**b**) If the system receives long signals with brief pauses ($T_{on} = 200\Delta t > \Delta T = 50\Delta t$), instead, the habituation mechanism promotes information storage and thus a reduction in the readout activity. The dashed black line indicates the corresponding Pareto front. Simulation parameters are as in *Appendix 1—table 1*.

## Appendix 3

### The necessity of storage

Here, we discuss in detail the necessity of slow storage implementing the negative feedback to have habituation. We will first investigate the possibility that negative feedback, necessary for any kind of habituative behaviors, is implemented directly through the readout population that undergoes a fast dynamics. We will analytically show that this limit leads to the absence of habituation, hinting at the necessity of having a slow dynamical feedback in the system (Sec. 1). Then, we will study the system in the scenario in which $U$ applies the feedback, bypassing the storage $S$, but it acts as a slow variable. Solving the Master Equation through our iterative numerical method, we show that, also in this case, habituation disappears (Sec. E). These results suggest that not only the feedback must be applied by a slow variable, but that such a slow variable must have a role different from the readout population, in line with recent observations in neural systems (*Fotowat and Engert, 2023*). The model proposed in the main text is indeed minimal in this respect, other than compatible with biological examples.

### Dynamical feedback cannot be implemented by a fast readout

If the storage is directly implemented by the readout population, the transition rates get modified as follows:

$$\begin{aligned}
\Gamma_{P\to A}^{(H)} &= e^{\beta(h-\Delta E)}\Gamma_R^0 & \Gamma_{A\to P}^{(H)} &= \Gamma_R^0 \\
\Gamma_{P\to A}^{(C)} &= e^{-\beta\Delta E}\Gamma_R^0 & \Gamma_{A\to P}^{(C)} &= e^{\beta\theta u}\Gamma_R^0 \\
\Gamma_{u\to u+1} &= e^{-\beta(V-cr)}\Gamma_U^0 & \Gamma_{u+1\to u} &= \Gamma_U^0(u+1)
\end{aligned} \tag{A3 - 1}$$

At this level, $\theta$ is a free parameter playing the same role as $\kappa/N_S$ in the complete model with the storage. We start again from the master equation for the propagator $P(u,r,h,t|u_0,r_0,h_0,t_0)$:

$$\partial_t P = \left[\frac{\hat{W}_U(r)}{\tau_U} + \frac{\hat{W}_R(u,h)}{\tau_R} + \frac{\hat{W}_H}{\tau_H}\right]P, \tag{A3 - 2}$$

where $\tau_U \ll \tau_R \ll \tau_H$, since we are assuming, as before, that $U$ is the fastest variable. Here, $\epsilon = \tau_U/\tau_R$ and $\delta = \tau_R/\tau_H$. Notice that now $\hat{W}_R$ depends also on $u$. We can solve the system again by resorting to a timescale separation and scaling the time by the slowest timescale, $\tau_H$. We have:

$$\partial_t P = \left[\epsilon^{-1}\delta^{-1}\hat{W}_U(r) + \delta^{-1}\hat{W}_R(u,h) + \hat{W}_H\right]P. \tag{A3 - 3}$$

We now expand the propagator at first order in $\epsilon$, $P = P^{(0)} + \epsilon P^{(1)}$. Then, the order $\epsilon^{-1}$ of the master equation gives, as above, $P^{(0)} = p_{U|R}^{\text{st}}(u|r)\Pi(r,h,t|r_0,h_0,t_0)$. At order $\epsilon^0$, (*Equation A3-4*) leads to

$$\partial_t \Pi(r,h,t|r_0,h_0,t_0) = \left[\delta^{-1}\sum_u \hat{W}_R(u,h)p_{U|R}^{\text{st}}(u|r) + \hat{W}_H\right]\Pi(r,h,t|r_0,h_0,t_0). \tag{A3-4}$$

To solve this, we expand the propagator as $\Pi = \Pi^{(0)} + \delta\Pi^{(1)}$ and, at order $\delta^{-1}$, we obtain:

$$\left(\sum_u \hat{W}_R(u,h)p_{U|R}^{\text{st}}(u|r)\right)\Pi^{(0)} = 0 \tag{A3 - 5}$$

This is a $2\times 2$ effective matrix acting on $\Pi^{(0)}$, where the only rate affected by $u$ is $\Gamma_{A\to P}^{(C)}$, which multiplies the active states, that is $r = 1$. This equation can be analytically computed, and the solution of *Equation A3-6* is:

$$\Pi^{(0)} = \rho_{R|H}^{\text{st}}(r|h)f(h,t|h_0 t_0) \qquad \rho_{R|H}^{\text{st}}(r=0|h) = \frac{e^{\beta\Delta E}(1+\Theta)}{e^{\beta h}+1+e^{\beta\Delta E}(1+\Theta)} \tag{A3-6}$$

with $\log(\Theta) = e^{-\beta(V-c)}(e^{\beta\theta}-1)$. Clearly, $\rho_{R|H}^{\text{st}}(r|h)$ does not depend on $u$ since we summed over the fast variable. Going on with the computation, at order $\delta^0$, we obtain:

$$\partial_t f(h, t | h_0, t_0) = \hat{W}_H f(h, t | h_0, t_0) \tag{A3 - 7}$$

So that the full propagator results to be:

$$P^{(0)}(u, r, h, t | u_0, r_0, h_0, t_0) = p_{U|R}^{\text{st}}(u | r) \rho_{R|H}^{\text{st}}(r | h) P_H(h, t | h_0, t_0) \tag{A3 - 8}$$

From this expression, we can find the joint probability distribution, following the same steps as before:

$$p_{U,R,H}(u, r, h, t) = p_{U|R}^{\text{st}}(u | r) \rho_{R|H}^{\text{st}}(r | h) p_H(h, t) \tag{A3 - 9}$$

As expected, since $U$ relaxes instantaneously, the feedback is instantaneous as well. As a consequence, the time-dependent behavior of the system is solely driven by the external signal $H$, with a fixed amplitude that takes into account the effect of the feedback only on average. This means that there will be no dynamic reduction of activity and, as such, no habituation in this scenario. This was somehow expected, since all variables are faster than the external signal and, as a consequence, the feedback cannot be implemented over time. The first conclusion is that the variable implementing the feedback has to evolve together with $H$.

## Effective dynamical feedback requires an additional population

We now assume that the feedback is, again, implemented by $U$, but it acts as a slow variable. Formally, we take $\tau_R \ll \tau_U \approx \tau_H$. Rescaling the time by the slowest timescale, $\tau_H$ (works the same for $\tau_U$), we have:

$$\partial_t P = \left[ \frac{\tau_H}{\tau_U} \hat{W}_U(r) + \epsilon^{-1} \hat{W}_R(u, h) + \hat{W}_H \right] P \tag{A3 - 10}$$

with $\epsilon = \tau_R / \tau_H$. We now expand the propagator at first order in $\epsilon$, $P = P^{(0)} + \epsilon P^{(1)}$. Then, the order $\epsilon^{-1}$ of the master equation is simply $\hat{W}_R P^{(0)} = 0$, whose solution gives $P^{(0)} = p_{R|U,H}^{\text{st}}(r | u, h) \Pi(u, h, t | u_0, h_0, t_0)$. At order $\epsilon^0$:

$$\partial_t \Pi(u, h, t | u_0, h_0, t_0) = \left[ \frac{\tau_H}{\tau_U} \sum_r \hat{W}_U(r) p_{R|U,H}^{\text{st}}(r | u, h) + \hat{W}_H \right] \Pi(u, h, t | u_0, h_0, t_0). \tag{A3 - 11}$$

The only dependence on $r$ in $\hat{W}_U(r)$ is through the production rate of $U$. Indeed, the effective transition matrix governing the birth-and-death process of readout molecules is characterized by:

$$\Gamma_{u \to u+1}^{\text{eff}} = e^{-\beta V} \left( e^{\beta c} p_{R|U,H}^{\text{st}}(r = 1 | u, h) + p_{R|U,H}^{\text{st}}(r = 0 | u, h) \right) \Gamma_U^0 \tag{A3 - 12}$$

This rate depends only on $h$, but $h$ evolves in time. Therefore, we should scan all possible (infinite) values that $h$ takes and build an infinite-dimensional transition matrix. In order to solve the system, imagine that we are looking at the interval $[t_0, t_0 + \Delta t]$. Then, we can employ the following approximation if $\Delta t \ll \tau_H$:

$$\Gamma_{u \to u+1}^{\text{eff}}(h) = \Gamma_{u \to u+1}^{\text{eff}}(h_0) \tag{A3 - 13}$$

Using this simplification, we need to solve the following equation:

$$\partial_t \Pi(u, h, t_0 + \Delta t | u_0, h_0, t_0) = \left[ \frac{\tau_H}{\tau_U} \hat{W}_U^{\text{eff}}(u, h_0) + \hat{W}_H \right] \Pi(u, h, t_0 + \Delta t | u_0, h_0, t_0). \tag{A3 - 14}$$

The explicit solution in the interval $t \in [t_0, t_0 + \Delta t]$ can be found to be:

$$\Pi(u, h, t_0 + \Delta t | u_0, h_0, t_0) = P_U^{\text{eff}}(u, t_0 + \Delta t | u_0, h_0, t_0) P_H(h, t_0 + \Delta t | h_0, t_0) \tag{A3 - 15}$$

with $P_U^{\text{eff}}$ a propagator. The full propagator at time $t_0 + \Delta t$ is then:

$$p_{U,R,H}(u, r, h, t_0 + \Delta t | u_0, r_0, h_0, t_0) =$$

$$\sum_{u_0} p_{R|U,H}^{\text{st}}(r|u, h) P_U^{\text{eff}}(u, t_0 + \Delta t | u_0, h_0, t_0) P_H(h, t_0 + \Delta t | h_0, t_0) p_{U,H}(u_0, h_0, t_0) \quad \text{(A3 - 16)}$$

Integrating over the initial conditions, we finally obtain:

$$p_{U,R,H}(u, r, h, t_0 + \Delta t) =$$

$$\sum_{u_0} p_{R|U,H}^{\text{st}}(r|u, h) \int dh_0 P_U^{\text{eff}}(u, t_0 + \Delta t | u_0, h_0, t_0) P_H(h, t_0 + \Delta t | h_0, t_0) p_{U,H}(u_0, h_0, t_0) \quad \text{(A3 - 17)}$$

To numerically integrate this equation, we make two approximations. The first one is that we solve the dynamics in all intervals in which the signal does not evolve, where $P_H$ is a delta function peaked at the initial condition. For all time points in which the signal changes, this amounts to considering the signal at the previous instant, a good approximation as long $\Delta t \ll \tau_H$, particularly when the time dependence of the signal is a square wave, as in our case.

The second approximation is to compute the propagator of $P_U$. As explained in the Materials and methods of the main text, we restrict our computation to the transitions between $n$ nearest neighbors in the $U$ space. In the case of transitions only among next-nearest neighbors, we have the following dynamics:

$$\partial_t P(u|u_0, h) = W^{\text{nn}} P(u|u_0) \quad \text{(A3 - 18)}$$

with the transition matrix:

$$W_{12}^{\text{nn}} = \hat{W}_{u_0 \to u_0 - 1} = \Gamma_0^U u_0 \qquad W_{13}^{\text{nn}} = \hat{W}_{u_0 + 1 \to u_0 - 1} = 0$$
$$W_{21}^{\text{nn}} = \hat{W}_{u_0 - 1 \to u_0} = \Gamma_{u_0 \to u_0 - u_0}^{\text{eff}} \qquad W_{13}^{\text{nn}} = \hat{W}_{u_0 + 1 \to u_0} = \Gamma_0^U (u_0 + 1)$$
$$W_{32}^{\text{nn}} = \hat{W}_{u_0 + 1 \to u_0} = 0 \qquad W_{32}^{\text{nn}} = \hat{W}_{u_0 \to u_0 + 1} = \Gamma_{u_0 \to u_0 + 1}^{\text{eff}}$$

the diagonal is fixed to satisfy the conservation of normalization, as usual. The solution is:

$$P(u|u_0, h) = p_{U|H}^{\text{st}} + \sum_\nu w_\nu a^{(\nu)} e^{\lambda_\nu \Delta t} \quad \text{(A3 - 19)}$$

where $w_\nu$ and $\lambda_\nu$ are respectively eigenvectors and eigenvalues of the transition matrix $W^{\text{nn}}$. The coefficients $a^{(\nu)}$ have to be evaluated according to the condition at time $t_0$:

$$P_{U|H}(u|u_0, h) = p_{U|H}^{\text{st}} + \sum_\nu w_\nu a^{(\nu)} = \delta_{u,u_0} \quad \text{(A3 - 20)}$$

$\delta_{u,u_0}$ where is the Kronecker's delta. To evaluate the information content of this model, we also need:

$$p_U(u, t_0 + \Delta t) = \sum_{u_0} p_U(u_0, t_0) \int dh P_U^{\text{eff}}(u, t_0 + \Delta t | u_0, h, t_0) p_H(h, t_0 + \Delta t)$$

$$p_{U|H}(u, t_0 + \Delta t | h) = \sum_{u_0} P_U^{\text{eff}}(u, t_0 + \Delta t | u_0, h, t_0) P_U(u_0, t_0) \quad \text{(A3 - 21)}$$

In **Appendix 3—figure 1** we show that, in this model, $U$ does not display habituation. Rather, it increases upon repeated stimuli, acting as the storage in the main text. On the other hand, the probability of the receptor being active does habituate. This suggests that habituation can only occur in fast variables modulated by slow variables.

It is straightforward to intuitively understand why a direct feedback from $U$, with this population undergoing a slow dynamics, cannot lead to habituation. Indeed, at a fixed distribution of the external signal, the stationary solution of $\langle U \rangle$ already takes into account the effect of the negative feedback. Hence, if the system starts with a very low readout population (no signal), the dynamics induced by a switching signal can only bring $\langle U \rangle$ to its steady state with intervals in which the population will grow and intervals in which it decreases. Naively speaking, the dynamics of $\langle U \rangle$ becomes similar to the one

of the storage in the complete model, since it is actually playing the same role of storing information in this simplified context.

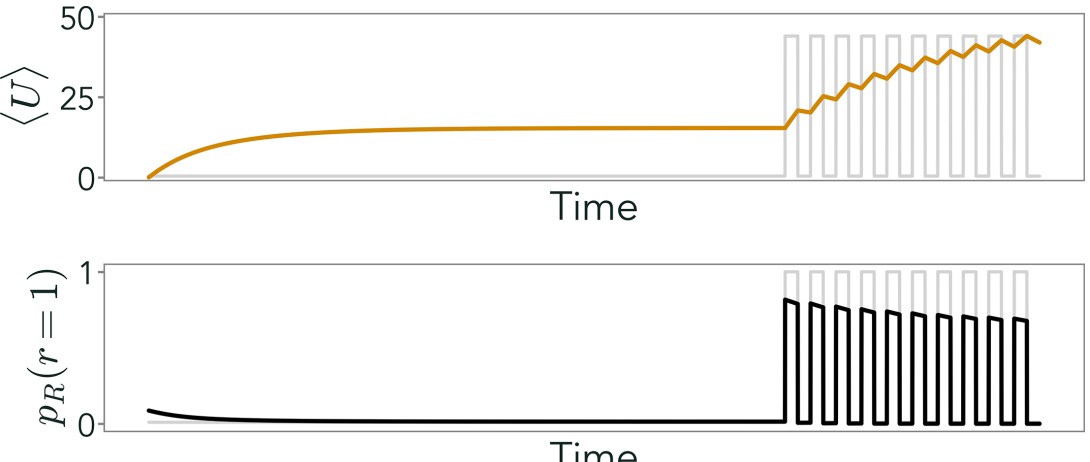

**Appendix 3—figure 1.** Dynamics of a system where $U$ evolves on the same timescale of $H$ and implements directly a negative feedback on the receptor. In this model, $\langle U \rangle$ (in red) increases upon repeated stimulation rather than decreasing, responding to changes in $\langle H \rangle$ (in gray) as the storage of the full model. On the other hand, the probability of the receptor being active, $pR(r = 1)$ (black), shows signs of habituation.

## Appendix 4

### Experimental setup

Acquisitions of the zebrafish brain activity were carried out in Elavl3:H2BGCaMP6s larvae at 5 days post fertilization raised at $28°C$ on a 12 hr light/12 hr dark cycle according to the approval by the Ethical Committee of the University of Padua (61/2020 dal Maschio). Larvae were embedded in 2% agarose gel and their brain activity was recorded using a multiphoton system with a custom 3D volumetric acquisition module. Briefly, the imaging path is based on an 8 kHz galvo-resonant commercial 2 P design (Bergamo I Series, Thorlabs, Newton, NJ, United States) coupled to a Ti:Sapphire source (Chameleon Ultra II, Coherent) tuned to 920 nm for imaging GCaMP6 signals and modulated by a Pockels cell (Conoptics). The fluorescence collection path includes a 705 nm long-pass main dichroic and a 495 nm long-pass dichroic mirror transmitting the fluorescence light toward a GaAsP PMT detector (H7422PA-40, Hamamatsu) equipped with EM525/50 emission filter. Data were acquired at 30 frames per second, using a water dipping Nikon CFI75 LWD 16 X W objective covering an effective field of view of about $450 \times 900 \, um$ with a resolution of 512 × 1024 pixels. The volumetric module is based on an electrically tunable lens (Optotune) moving continuously according to a saw-tooth waveform synchronized with the frame acquisition trigger. An entire volume of about $180 - 200 \, um$ in thickness encompassing 30 planes separated by about $7um$ is acquired at a rate of 1 volume per second, sufficient to track the relative slow dynamics associated with the fluorescence-based activity reporter GCaMP6s.

As for the visual stimulation, looming stimuli were generated using Stytra and presented monocularly on a $50 \times 50 \, mm$ screen using a DPL4500 projector by Texas Instruments. The dark looming dot was presented 10 times with 150 s interval, centered with the fish eye and with a l/v parameter of 8.3 s, reaching at the end of the stimulation a visual angle of $79.4°$ corresponding to an angular expansion rate of $9.5°/s$. The acquired temporal series were first processed using an automatic pipeline, including motion artifact correction, temporal filtering with a rectangular window 3 s long, and automatic segmentation using Suite2P. Then, the obtained dataset was manually curated to resolve segmentation errors or to integrate cells not detected automatically. We fit the activity profiles of about 52,000 cells with a linear regression model (scikit-learn Python Library) using a set of base functions representing the expected responses to each of the stimulation events, obtained by convolving an exponentially decaying kernel of the GCaMP signal lifetime with square waveforms characterized by an amplitude different from zero only during the presentation of the corresponding visual stimulus. The resulting coefficients were divided for the mean squared error of the fit to obtain a set of scores. The cells, whose score fell within the top 5 of the distribution, were considered for the dimensionality reduction analysis.

The resulting fluorescence signals $F^{(i)}$, for $i = 1, \ldots, N_{\text{cells}}$, were processed by removing a moving baseline to account for baseline drifting and fast oscillatory noise (*Jia et al., 2011*). Briefly, for each time point $t$, we selected a window $[t - \tau_2, t]$ and evaluated the minimum smoothed fluorescence,

$$F_0^{(i)} = \min_{u \in [t - \tau_2, t]} \left[ \frac{1}{\tau_1} \int_{u - \tau_1/2}^{u + \tau_1/2} F(s) \, ds \right]. \tag{A4 - 1}$$

Then, the relative change in fluorescence signal,

$$R^{(i)}(t) = \frac{F^{(i)}(t) - F_0^{(i)}}{F_0^{(i)}} \tag{A4 - 2}$$

is smoothed with an exponential moving average. Thus, the neural activity profile for the $i$-th cell that we use in the main text is given by

$$x^{(i)}(t) = \frac{\int_0^t R(t - \tau) w(\tau) d\tau}{\int_0^t w(\tau) d\tau}, \quad w(t) = \exp\left[ -\frac{t}{\tau_0} \right]. \tag{A4 - 3}$$

In accordance with the previous literature (*Jia et al., 2011*), we set $\tau_0 = 0.2 \, s$, $\tau_1 = 0.75 \, s$, and $\tau_2 = 3 \, s$. The qualitative nature of the low-dimensional activity in the PCA space is not altered by other sensible choices of these parameters.

