## [Editor Report · eLife Assessment]

This manuscript presents a **valuable** minimal model of habituation which is quantified by information theoretic measures. The results here could be of use in interpreting habituation behavior in a range of biological systems. The evidence presented is **solid**, and uses simulations of the minimal model to recapitulate several hallmarks of habituation from a simple model.

---

## [Referee Report · Reviewer #2 (Public review)]

In this study, the authors aim to investigate habituation, the phenomenon of increasing reduction in activity following repeated stimuli, in the context of its information theoretic advantage. To this end, they consider a highly simplified three-species reaction network where habituation is encoded by a slow memory variable that suppresses the receptor and therefore the readout activity. Using analytical and numerical methods, they show that in their model the information gain, the difference between the mutual information between the signal and readout after and before habituation, is maximal for intermediate habituation strength. Furthermore, they demonstrate that the Pareto front corresponding to an optimization strategy that maximizes the mutual information between signal and readout in the steady-state and minimizes dissipation in the system also exhibits similar intermediate habituation strength. Finally, they briefly compare predictions of their model to whole-brain recordings of zebrafish larvae under visual stimulation.

The author's simplified model serves as a good starting point for understanding habituation in different biological contexts as the model is simple enough to allow for some analytic understanding but at the same time exhibits most basic properties of habituation in sensory systems. Furthermore, the author's finding of maximal information gain for intermediate habituation strength via an optimization principle is, in general, interesting. However, the following points remain unclear:

(1) How general is their finding that the optimal Pareto front coincides with the region of maximal information gain? For instance, what happens if the signal H_st (H_max) isn't very strong? Does it matter that in this case, H_st only has a minor influence on delta Q_R? In the binary switching case, what happens if H_max is rather different from H_st (and not just 20% off)? Or in a case where the adapted value corresponds to the average of H_max and H_min?

(2) The comparison to experimental data isn't very convincing. For instance, is PCA performed simultaneously on both the experimental data set and on the model or separately? What are the units of the PCs in Fig. 6(b,c)? Given that the model parameters are chosen so that the activity decrease in the model is similar to the one in the data (i.e., that they show similar habituation in terms of the readout), isn't it expected that the dynamics in the PC1/2 space look very similar?

---

## [Referee Report · Reviewer #3 (Public review)]

The authors use a generic model framework to study the emergence of habituation and its functional role from information-theoretic and energetic perspectives. Their model features a receptor, readout molecules, and a storage unit, and as such, can be applied to a wide range of biological systems. Through theoretical studies, the authors find that habituation (reduction in average activity) upon exposure to repeated stimuli should occur at intermediate degrees to achieve maximal information gain. Parameter regimes that enable these properties also result in low dissipation, suggesting that intermediate habituation is advantageous both energetically and for the purpose of retaining information about the environment.

A major strength of the work is the generality of the studied model. The presence of three units (receptor, readout, storage) operating at different time scales and executing negative feedback can be found in many domains of biology, with representative examples well discussed by the authors (e.g. Figure 1b). A key takeaway demonstrated by the authors that has wide relevance is that large information gain and large habituation cannot be attained simultaneously. When energetic considerations are accounted for, large information gain and intermediate habituation appear to be the favorable combination.

Comments on the revision:

The authors have adequately addressed the points I raised during the initial review. The text has been clarified at multiple instances, and the treatment of energy expenditure is now more rigorous. The manuscript is much improved both in terms of readability and scientific content.

---

## [Author Response]

The following is the authors’ response to the original reviews

**Reviewer #1 (Public Review):**
Summary:The manuscript by Nicoletti et al. presents a minimal model of habituation, a basic form of non-associative learning, addressing both from dynamical and information theory aspects of how habituation can be realized. The authors identify that negative feedback provided with a slow storage mechanism is sufficient to explain habituation.Strengths:The authors combine the identification of the dynamical mechanism with information-theoretic measures to determine the onset of habituation and provide a description of how the system can gain maximum information about the environment.

We thank the reviewer for highlighting the strength of our work and for their comments, which we believe have been instrumental in significantly improving our work and its scope. Below, we address all their concerns.

Weaknesses:I have several main concerns/questions about the proposed model for habituation and its plausibility. In general, habituation does not only refer to a decrease in the responsiveness upon repeated stimulation but as Thompson and Spencer discussed in Psych. Rev. 73, 16-43 (1966), there are 10 main characteristics of habituation, including (i) spontaneous recovery when the stimulus is withheld after response decrement; dependence on the frequency of stimulation such that (ii) more frequent stimulation results in more rapid and/or more pronounced response decrement and more rapid spontaneous recovery; (iii) within a stimulus modality, the less intense the stimulus, the more rapid and/or more pronounced the behavioral response decrement; (iv) the effects of repeated stimulation may continue to accumulate even after the response has reached an asymptotic level (which may or may not be zero, or no response). This effect of stimulation beyond asymptotic levels can alter subsequent behavior, for example, by delaying the onset of spontaneous recovery.These are only a subset of the conditions that have been experimentally observed and therefore a mechanistic model of habituation, in my understanding, should capture the majority of these features and/or discuss the absence of such features from the proposed model.

We are really grateful to the reviewer for pointing out these aspects of habituation that we overlooked in the previous version of our manuscript. Indeed, our model is able to capture most of these 10 observed behaviors, specifically: (1) habituation; (2) spontaneous recovery; (3) potentiation of habituation; (4) frequency sensitivity; (5) intensity sensitivity; (6) subliminal accumulation. Here, we are following the same terminology employed in Eckert et al., Current Biology 34, 5646–5658 (2024), the paper highlighted by the reviewer. We have dedicated a section of the revised version of the manuscript to these hallmarks, substantiating the validity of our framework as a minimal model to have habituation. We remark that these are the sole hallmarks that can be discussed by considering one single external stimulus and that can be identified without ambiguity in a biochemical context. This observation is again in line with Eckert et al., Current Biology 34, 5646–5658 (2024).

In the revised version, we employ the same strategy of the aforementioned work to determine when the system can be considered “habituated”. Indeed, we introduce a response threshold that is now discussed in the manuscript. We also included a note in the discussions stating that, since any biochemical model will eventually reach a steady state, subliminal accumulation, for example, can only be seen with the use of a threshold. The introduction of different storage mechanisms, ideally more detailed at a molecular level, can shed light on this conceptual gap. This is an interesting direction of research.

Furthermore, the habituated response in steady-state is approximately 20% less than the initial response, which seems to be achieved already after 3-4 pulses, the subsequent change in response amplitude seems to be negligible, although the authors however state "after a large number of inputs, the system reaches a time-periodic steady-state". How do the authors justify these minimal decreases in the response amplitude? Does this come from the model parametrization and is there a parameter range where more pronounced habituation responses can be observed?

The reviewer is correct, but this is solely a consequence of the specific set of parameters we selected. We made this choice solely for visualization purposes in the previous version. In the revised version, in the section discussing the hallmarks of habituation, we also show other parameter choices when the response decrement is more pronounced. Moreover, we remark that the contour plot of \Delta⟨U> clearly shows that the decrement can largely exceed the 20% threshold presented in the previous version.

In the revised version, also in light of the works highlighted by the reviewer, we decided to move the focus of the manuscript to the information-theoretic advantage of habituation. As such, we modified several parts of the main text. Also, in the region of optimal information gain, habituation is at an intermediate level. For this reason, we decided to keep the same parameter choice as the previous version in Figure 2.

We stated that the time-periodic steady-state is reached “after a large number of stimuli” from a mathematical perspective. However, by using a habituation threshold, as done in Eckert et al., Current Biology 34, 5646–5658 (2024), we can state that the system is habituated after a few stimuli for each set of parameters. This aspect is highlighted in the revised version of the manuscript (see also the point above).

The same is true for the information content (Figure 2f) - already at the first pulse, IU, H ~ 0.7 and only negligibly increases afterwards. In my understanding, during learning, the mutual information between the input and the internal state increases over time and the system extracts from these predictions about its responses. In the model presented by the authors, it seems the system already carries information about the environment which hardly changes with repeated stimulus presentation. The complexity of the signal is also limited, and it is very hard to clarify from the presented results, whether the proposed model can actually explain basic features of habituation, as mentioned above.

As for the response decrement of the readout, we can certainly choose a set of parameters for which the information gain is higher. In the revised version, we also report the information at the first stimulation and when the system is habituated to give a better idea of the range of these quantities. At any rate, as the referee correctly points out, it is difficult to give an intuitive interpretation of the information in our minimal model.

It is also important to remark that, since the readout population and the receptor both undergo fast dynamics (with appropriate timescales as discussed in the text), we are not observing the transient gain of information associated with the first stimulus. As such, the mutual information presents a discontinuous behavior that resembles the dynamics of the readout, thereby starting at a non-zero value already at the first stimulus.

Additionally, there have been two recent models on habituation and I strongly suggest that the authors discuss their work in relation to recent works (bioRxiv 2024.08.04.606534; arXiv:2407.18204).

We thank the reviewer for pointing out these relevant references. In the revised version, we highlighted that we discuss the information-theoretic aspects of habituation, while the aforementioned references focus on the dynamics of this phenomenon.

**Reviewer #1 (Recommendations for the authors):**
I would also like to note here the simplification of the proposed biological model - in particular, that the receptor can be in an active/passive state, as well as proposing the Nf-kB signaling module as a possible molecular realization. Generally, a large number of cell surface receptors including RTKs of GPCRs have much more complex dynamics including autocatalytic activation that generally leads to bistability, and the Nf-kB has been demonstrated to have oscillatory even chaotic dynamics (works of Savas Tsay, Mogens Jensen and others). Considering this, the authors should at least discuss under which conditions these TNF-Alpha signaling could potentially serve as a molecular realisation for habituation.

We thank the reviewer for bringing this to our attention. In the previous version, we reported the TNF signaling network only to show a similar coarse-grained modular structure. However, following a suggestion of reviewer #2, we decided to change Figure 1 to include a simplified molecular scheme of chemotaxis rather than TNF signaling, to avoid any source of confusion about this issue.

Also, a minor point: Figures 2d-e are cited before 2a-c.

We apologize for the oversight. The structure of the Figures and their order is now significantly different, and they are now cited in the correct order.

**Reviewer #2 (Public review):**
In this study, the authors aim to investigate habituation, the phenomenon of increasing reduction in activity following repeated stimuli, in the context of its information-theoretic advantage. To this end, they consider a highly simplified three-species reaction network where habituation is encoded by a slow memory variable that suppresses the receptor and therefore the readout activity. Using analytical and numerical methods, they show that in their model the information gain, the difference between the mutual information between the signal and readout after and before habituation, is maximal for intermediate habituation strength. Furthermore, they demonstrate that the Pareto front corresponds to an optimization strategy that maximizes the mutual information between signal and readout in the steady state, minimizes some form of dissipation, and also exhibits similar intermediate habituation strength. Finally, they briefly compare predictions of their model to whole-brain recordings of zebrafish larvae under visual stimulation.The author's simplified model might serve as a solid starting point for understanding habituation in different biological contexts as the model is simple enough to allow for some analytic understanding but at the same time exhibits all basic properties of habituation in sensory systems. Furthermore, the author's finding of maximal information gain for intermediate habituation strength via an optimization principle is, in general, interesting. However, the following points remain unclear or are weakly explained:

We thank the reviewer for deeming our work interesting and for considering it a solid starting point for understanding habituation in biological systems.

(1) Is it unclear what the meaning of the finding of maximal information gain for intermediate habituation strength is for biological systems? Why is information gain as defined in the paper a relevant quantity for an organism/cell? For instance, why is a system with low mutual information after the first stimulus and intermediate mutual information after habituation better than one with consistently intermediate mutual information? Or, in other words, couldn't the system try to maximize the mutual information acquired over the whole time series, e.g., the time series mutual information between the stimulus and readout?

This is a delicate aspect to discuss and we thank the referee for the comment. In the revised version, we report information gain, initial and final information, highlighting that both gain and final information are higher in regions where habituation is present. They have qualitatively similar behavior and highlight a clear information-theoretic advantage of this dynamical phenomenon. An important point is that, to determine the optimal Pareto front, we consider a prolonged stimulus and its associated steady-state information. Therefore, from the optimization point of view, there is no notion of “information gain” or “final information”, which are intrinsically dynamical quantities. As a result, the fact that optimal curve lies in the region of optimal information gain is a-priori not expected and hints at the potential crucial role of this feature. In the revised version, we elucidate this aspect with several additional analyses.

We would like to add that, from a naive perspective, while the first stimulation will necessarily trigger a certain (non-zero) mutual information, multiple observations of the same stimulus have to reflect into accumulated information that consequently drives the onset of observed dynamical behaviors, such as habituation.

(2) The model is very similar to (or a simplification of previous models) for adaptation in living systems, e.g., for adaptation in chemotaxis via activity-dependent methylation and demethylation. This should be made clearer.

We apologize for having missed this point. Our choice has been motivated by the fact that we wanted to avoid confusion between the usual definition of (perfect) adaptation and habituation. However, we now believe that this is not the case for the revised manuscript, and we now include chemotaxis as an example in Figure 1.

(3) It remains unclear why this optimization principle is the most relevant one. While it makes sense to maximize the mutual information between stimulus and readout, there are various choices for what kind of dissipation is minimized. Why was \begin{document}$\delta Q_R$\end{document} chosen and not, for instance, \begin{document}$\dot{\Sigma}_{int}$\end{document} or the sum of both? How would the results change in that case? And how different are the results if the mutual information is not calculated for the strong stimulation input statistics but for the background one?

We thank the reviewer for the suggestion. We agree that a priori, there is no reason to choose \delta Q_R or a function of the internal energy flux J_int (that, in the revised version, we are using in place of \dot\Sigma_int following the suggestion of reviewer #3). The rationale was to minimize \delta Q_R since this dissipation is unavoidable and stems from the presence of the storage inhibiting the receptor through the internal pathway. Indeed, considering the existence of two different pathways implementing sensing and feedback, the presence of any input will result in a dissipation produced by the receptor. This energy consumption is reflected in \delta Q_R.

In the revised version, we now include in the optimization principle two energy contributions (see Eq. (14) of the revised manuscript): \delta Q_R and E_int, which is the energy consumption associated with the driven storage production per unit energy. All Figures have been updated accordingly. The results remain similar, as \delta Q_R still represents the main contribution, especially at high \beta.

Furthermore, in the revised version, we include examples of the Pareto optimization for different values of input strength. As detailed both in the main text and the Supplementary Information, changing the value of ⟨H⟩ moves the Pareto frontier in the (\beta, \sigma) space, since the signal needs to be strong enough for the system to distinguish it from the intrinsic thermal noise (controlled by beta). We also show that if the system is able to tune the inhibition strength \kappa, the Pareto frontiers at different ⟨H⟩ collapse into a single curve. This shows that, although the values of, e.g., the mutual information, depend on ⟨H⟩, the qualitative behavior of the system in this regime is effectively independent of it. We also added more details about this in the Supplementary Information.

(4) The comparison to the experimental data is not too strong of an argument in favor of the model. Is the agreement between the model and the experimental data surprising? What other behavior in the PCA space could one have expected in the data? Shouldn't the 1st PC mostly reflect the "features", by construction, and other variability should be due to progressively reduced activity levels?

The agreement between data and model is not surprising - we agree on this - since the data exhibit habituation. However, we believe that the fact that our minimal model is able to capture the features of a complex neural system just by looking at the PCs, without any explicit biological details, is non-trivial. We also stress that the 1st PC only reflects the feature that captures most of the variance of the data and, as such, it is difficult to have a-priori expectations on what it should represent. In the case of the data generated from the model, most of the variance of the activity comes from the switching signal, and similar considerations can be made for the looming stimulations in the data. We updated the manuscript to clarify this point.

**Reviewer #2 (Recommendations for the authors):**
(1) The abstract makes it sound like a new finding is that habituation is due to a slow, negative feedback mechanism. But, as mentioned in the introduction, this is a well-known fact.

We agree with the reviewer. We have revised the abstract.

(2) Figure 2c Why does the range of Delta Delta I_f include negative values if the corresponding region is shaded (right-tilted stripes)?

The negative values in the range are those attained in the shaded region with right-tilted stripes. We decided to include them in the colorbar for clarity, since Delta Delta I_f is also plotted in the region where it attains negative values.

(3) What does the Pareto front look like if the optimization is done for input statistics given by ⟨H⟩_min?

In the revised version, we include examples of the Pareto optimization for different values of input strength. As detailed both in the main text and the Supplementary Information, changing the value of ⟨H⟩ moves the Pareto frontier in the (\beta, \sigma) space, since the strength of the signal is crucial for the system to discriminate input and thermal noise (see also the answers above).

In particular, in Figure 4 we explicitly compare the results of the Pareto optimization (which is done with a static input of a given statistics) with the dynamics of the model for different values of ⟨H⟩ in two scenarios, i.e., adaptive and non-adaptive inhibition strength (see answers above for details).

We also remark that ⟨H⟩_min represents the background signal that the system is not trying to capture, which is why we never used it for optimization.

(4) From the main text, it is rather difficult to understand how the comparison to the experimental data was performed. How was the PCA done exactly? What are the "features" of the evoked neural response?

The PCA on data is performed starting from the single-neuron calcium dynamics. To perform a far comparison, we reconstruct a similar but extremely simplified dynamics using our model as explained in Methods to perform the PCA on analogous simulated data. We added a comment on this in the revised version. While these components capture most of the variance in the data, their specific interpretation is usually out of reach and we believe that it lies beyond the scope of this theoretical work. We also remark that the model does not contain all these biological details - a strong aspect in our opinion - and, as such, it cannot capture specific biological features.

**Reviewer #3 (Public review):**
The authors use a generic model framework to study the emergence of habituation and its functional role from information-theoretic and energetic perspectives. Their model features a receptor, readout molecules, and a storage unit, and as such, can be applied to a wide range of biological systems. Through theoretical studies, the authors find that habituation (reduction in average activity) upon exposure to repeated stimuli should occur at intermediate degrees to achieve maximal information gain. Parameter regimes that enable these properties also result in low dissipation, suggesting that intermediate habituation is advantageous both energetically and for the purpose of retaining information about the environment.A major strength of the work is the generality of the studied model. The presence of three units (receptor, readout, storage) operating at different time scales and executing negative feedback can be found in many domains of biology, with representative examples well discussed by the authors (e.g. Figure 1b). A key takeaway demonstrated by the authors that has wide relevance is that large information gain and large habituation cannot be attained simultaneously. When energetic considerations are accounted for, large information gain and intermediate habituation appear to be a favorable combination.

We thank the reviewer for this positive assessment of our work and its generality.

While the generic approach of coarse-graining most biological detail is appealing and the results are of broad relevance, some aspects of the conducted studies, the problem setup, and the writing lack clarity and should be addressed:(1) The abstract can be further sharpened. Specifically, the "functional role" mentioned at the end can be made more explicit, as it was done in the second-to-last paragraph of the Introduction section ("its functional advantages in terms of information gain and energy dissipation"). In addition, the abstract mentions the testing against experimental measurements of neural responses but does not specify the main takeaways. I suggest the authors briefly describe the main conclusions of their experimental study in the abstract.

We thank the reviewer for raising this point. In the revised version, we have changed the abstract to reflect the reviewer’s points and the new structure and results of the manuscript.

(2) Several clarifications are needed on the treatment of energy dissipation.- When substituting the rates in Eq. (1) into the definition of δQ_R above Eq. (10), "σ" does not appear on the right-hand side. Does this mean that one of the rates in the lower pathway must include σ in its definition? Please clarify.

We apologize to the reviewer for this typo. Indeed, \sigma sets the energy scale of feedback and, as such, it appears in the energetic driving given by the feedback on the receptor, i.e., in Eq. (1) together with \kappa. This typo has been corrected in the revised manuscript, and all subsequent equations are consistent.

- I understand that the production of storage molecules has an associated cost σ and hence contributes to dissipation. The dependence of receptor dissipation on ⟨H⟩, however, is not fully clear. If the environment were static and the memory block was absent, the term with ⟨H⟩ would still contribute to dissipation. What would be the nature of this dissipation?

In the spirit of building a paradigmatic minimal model with a thermodynamic meaning, we considered H to act as an external thermodynamic driving. Since this driving acts on a different pathway with respect to the one affected by the storage, the receptor is driven out of equilibrium by its presence.

By eliminating the memory block, we would also be necessarily eliminating the presence of the pathway associated with the storage effect (“internal pathway” in the manuscript), since its presence is solely due to the existence of a storage population. Therefore, in this case, the receptor would be a 2-state, 1-pathway system and, as such, it would always satisfy an effective detailed balance. As a consequence, the definition of \delta Q_R reported in the manuscript would not hold anymore and the receptor would not exhibit any dissipation. Thus, in a static environment and without a memory block, no receptor dissipation would be present. We would also like to stress that our choice to model two different pathways has been motivated by the observation that the negative feedback acts along a different pathway in several biochemical and biological examples. We made some changes to the model description in the revised version and we hope that this aspect has been clarified.

- Similarly, in Eq. (9) the authors use the ratio of the rates Γ_{s → s+1} and Γ_{s+1 → s} in their expression for internal dissipation. The first-rate corresponds to the synthesis reaction of memory molecules, while the second corresponds to a degradation reaction. Since the second reaction is not the microscopic reverse of the first, what would be the physical interpretation of the log of their ratio? Since the authors already use σ as the energy cost per storage unit, why not use σ times the rate of producing S as a metric for the dissipation rate?

We agree with the referee that the reverse reaction we considered is not the microscopic reverse of the storage production. In the case of a fast readout population, we employed a coarse-grained view to compute this entropy production. To be more precise, we gladly welcomed the referee’s suggestion in the revised version and modified the manuscript accordingly. As suggested, we now employ the energy flux associated with the storage production to estimate the internal dissipation (see new Fig. 3).

In the revised version, we also use this quantity in the optimization procedure in combination with \deltaQ_R (see new Fig. 4) to have a complete characterization of the system’s energy consumption. The conclusions are qualitatively identical to before, but we believe that now they are more solid from a theoretical perspective. For this important advance in the robustness and quality of our work, we are profoundly grateful to the referee.

(3) Impact of the pre-stimulus state. The plots in Figure 2 suggest that the environment was static before the application of repeated stimuli. Can the authors comment on the impact of the pre-stimulus state on the degree of habituation and its optimality properties? Specifically, would the conclusions stay the same if the prior environment had stochastic but aperiodic dynamics?

The initial stimulus is indeed stochastic with an average constant in time and mimics the background (small) signal. We apply the (strong) stimulation when the system already reached a stationary state with respect to the background. As it can be appreciated in Fig. 2 of the revised version, the model response depends on the pre-stimulus level, since it sets the storage concentration before the stimulation arrives and, as such, the subsequent habituation dynamics. This dependence is important from a dynamical perspective. The information-theoretic picture has been developed, as said above, by letting the system relax before the first stimulus. This eliminates this arbitrary dependence and provides a clearer idea of the functional advantages of habituation. Moreover, the optimization procedure is performed in a completely different setting, with no pre-stimulus at all, since we only have one prolonged stimulation. We hope that the revised version is clearer on all these points.

(4) Clarification about the memory requirement for habituation. Figure 4 and the associated section argue for the essential role that the storage mechanism plays in habituation. Indeed, Figure 4a shows that the degree of habituation decreases with decreasing memory. The graph also shows that in the limit of vanishingly small Δ⟨S⟩, the system can still exhibit a finite degree of habituation. Can the authors explain this limiting behavior; specifically, why does habituation not vanish in the limit Δ⟨S⟩ -> 0?

We apologize for the lack of clarity and we thank the reviewer for spotting this issue. In Figure 4 (now Figure 5 in the revised manuscript) Δ⟨S⟩ is not exactly zero, but equal to 0.15% at the final point. It appeared as 0% in the plot due to an unwanted rounding in the plotting function that we missed. This has been fixed in the revised version, thank you.

**Reviewer #3 (Recommendations for the authors):**
(1) Page 2 | "Figure 1b-e" should be "Figure 1b-d" since there is no panel (e) in Figure 1.(2) Figure 1a | In the top schematic, the symbol "k" is used, while in the rest of the text, the proportionality constant is denoted by κ.

We thank the reviewer for pointing this out. Figure 1 has been revised and the panels are now consistent. The proportionality constant (the inhibition strength) has also been fixed.

(3) Figure 1a | I find the upper part of the schematic for Storage hard to perceive. I understand the lower part stands for the degradation reaction for storage molecules. The upper part stands for the synthesis reaction catalyzed by the readout population. I think the bolded upper arrow would explain it sufficiently well; the left/right arrows, together with the crossed green circle make that part of the figure confusing. Consider simplifying.

We decided to remove the left/right arrows, as suggested by the reviewer, as we agree that they were unnecessarily complicating the schematic. We hope that the revised version will be easier to understand.

(4)Page 3 | It would be helpful to tell what the temporal statistics of the input signal $p_H(h,t)$ is, i.e. <h(t) h(t')>. Looking at the example trajectory in Figure 1a, consecutive signal values do not seem correlated.

We agree with the reviewer that this is an important detail and worth mentioning. We now explicitly state that consecutive values are not correlated, for simplicity.

(5)Figure 2 | I believe the label "EXTERNAL INPUT" refers to the *average* external input, not one specific realization (similar to panels (d) and (e) that report on average metrics). I suggest you indicate this in the label, or, what may be even better, add one particular realization of the stochastic input to the same graph.

We thank the reviewer for spotting this. We now write that what we show is the average external signal. We prefer this solution rather than showing a realization of the stochastic input, since it is more consistent with the rest of the plots, where we always show average quantities. We also note that Figure 2 is now Figure 3 in the revised manuscript.

(6)Figure 2d | The expression of Δ⟨U⟩ is the negative of the definition in Eq. (5). It should be corrected.

In the revised version, both the definitions in Figure 2 (now Figure 3) and in the text (now Eq. (11)) are consistent.

(7) Figure 3(d-e) caption | "where ⟨U⟩ starts to be significantly smaller than zero." There, it should be Δ⟨U⟩ instead of ⟨U⟩.

Thanks again, we corrected this typo.